# CONTINUAL TRAFFIC FORECASTING VIA MIXTURE OF EXPERTS

## ABSTRACT

The real-world traffic networks undergo expansion through the installation of new sensors, implying that the traffic patterns continually evolve over time. Incrementally training a model on the newly added sensors would make the model forget the past knowledge, i.e., catastrophic forgetting, while retraining the model on the entire network to capture these changes is highly inefficient. To address these challenges, we propose a novel Traffic Forecasting Mixture of Experts (TFMoE) for traffic forecasting under evolving networks. The main idea is to segment the traffic flow into multiple homogeneous groups, and assign an expert model responsible for a specific group. This allows each expert model to concentrate on learning and adapting to a specific set of patterns, while minimizing interference between the experts during training, thereby preventing the dilution or replacement of prior knowledge, which is a major cause of catastrophic forgetting. Through extensive experiments on a real-world long-term streaming network dataset, PEMSD3-Stream, we demonstrate the effectiveness and efficiency of TFMoE. Our results showcase superior performance and resilience in the face of catastrophic forgetting, underscoring the effectiveness of our approach in dealing with continual learning for traffic flow forecasting in long-term streaming networks.

## 1 INTRODUCTION

Recently, numerous studies have been proposed with the goal of enhancing the accuracy of traffic forecasting. However, while various models have been proposed to tackle this problem (Li et al., 2017; Chen et al., 2021b; Zhang et al., 2018a; Yu et al., 2017; Wu et al., 2019; Fang et al., 2021; Park et al., 2020; Zheng et al., 2020; Guo et al., 2019; Shang et al., 2021; Cao et al., 2020; Bai et al., 2020; Li & Zhu, 2021; Lu et al., 2020; Lan et al., 2022), most of them focus on improving accuracy in *static* traffic networks.

In this work, we focus on the real-world traffic forecasting scenarios, where the traffic networks undergo expansion through the installation of new sensors in the surrounding areas (i.e., *evolving* traffic network). While these newly added sensors may exhibit traffic patterns similar to pre-existing ones, they also introduce previously unobserved patterns. Moreover, even pre-existing sensors may display new patterns over long-term periods (e.g., several years) due to various factors such as urban development, infrastructure projects, or alterations in traffic demand stemming from population migration and urban population growth. Consequently, if a model that is trained on the past traffic network is further incrementally trained on the newly added sensors in the expanded network, the model would forget the past knowledge, resulting in a severe performance degradation on the pre-existing sensors in the past network, which is called catastrophic forgetting. A straightforward solution would be to re-train the model on the entire dataset containing not only the newly added sensors but also the pre-existing sensors. However, the process of retraining the model is computationally demanding and time-consuming, highlighting the necessity for a more suitable and efficient learning methodology.

TrafficStream (Chen et al., 2021a) is a pioneering work that focuses on traffic forecasting under evolving traffic networks. Its main idea is to adopt continual learning strategies to continuously learn and adapt from ongoing data streams, integrating new information while preserving the past knowledge (i.e., avoid catastrophic forgetting). By utilizing popular methods in continual learning such as Elastic Weight Consolidation (EWC) (Kirkpatrick et al., 2017) and Replay (Robins, 1995; Rebuffi et al., 2017), TrafficStream manages the expansion and evolution of traffic networks. However, despite its effectiveness, the dynamic nature of traffic networks continues to pose significant

challenges, highlighting the necessity for a more adaptive approach to accommodate these diverse and evolving traffic patterns. Specifically, TrafficStream adopts a "one-model-fits-all" approach, which uses a single model to capture all the evolving traffic patterns, being susceptible to catastrophic forgetting.

To this end, we propose Traffic Forecasting Mixture of Experts (TFMoE) for traffic forecasting under evolving networks. The main idea of TFMoE is to segment the traffic data into multiple homogeneous groups, and assign an expert traffic forecasting model to each group. By doing so, we allow each model to be responsible for predicting the traffic flow of its assigned group, minimizing interference with each other during training, which in turn alleviates catastrophic forgetting. This is possible because each model can concentrate on learning and adapting to a specific set of patterns, thereby preventing the dilution or replacement of prior knowledge, which is a major cause of catastrophic forgetting. Figure 1 demonstrates the motivation of our work. We observe that the sensors can be segmented into multiple homogeneous groups based on their traffic patterns, and moreover the newly added sensors tend to belong to one of the existing clusters. That is, even if the traffic network is expanded, the newly added sensors exhibit similar traffic patterns as those of pre-existing sensors. This implies that having an expert solely dedicated to each homogeneous group would be more effective than the "one-model-fits-all" approach in terms of alleviating catastrophic forgetting. This is because each expert only needs to concentrate on learning and adapting to a specific set of patterns, while the "one-model-fits-all" approach needs to adapt to the global dynamics even though local changes mainly occur.

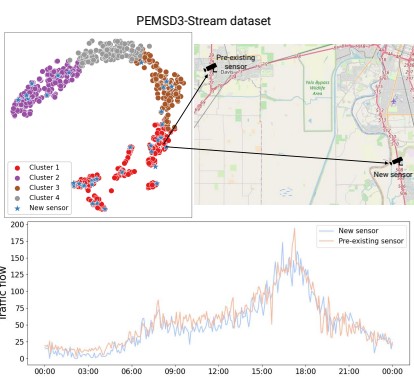

Figure 1: The top-left plot depicts the t-SNE visualization of one week's traffic patterns gathered from each sensor in the traffic network, with newly added sensors of the next year marked by stars. The top-right image shows the geographical location of a new sensor and its closest counterpart in the latent space. The bottom plot indicates the notable similarity between the traffic patterns obtained from these two sensors.

Each expert model in TFMoE contains two components: 1) a reconstructor that utilizes a Variational Autoencoder (VAE) structure to reconstruct the traffic flow, and 2) a predictor that makes future predictions based on the past traffic flow. We first cluster the traffic flow based on the representation extracted by a pre-trained feature extractor, and train an expert model (i.e., a reconstructor and a predictor) on each cluster. Then, when a new traffic flow is introduced, we assign it to an expert model whose reconstruction loss is the smallest. Finally, we make the final prediction by combining the individual predictions from each expert model using a reconstruction-based gating mechanism.

At the core of our approach are three pivotal strategies. 1) '*Reconstructor-Based Knowledge Consolidation Loss*,' inspired by the learning without forgetting technique, ensures that the model learns new traffic patterns while also preserving knowledge from previous tasks based on the concept of the localized group within the VAE. 2) '*Forgetting-Resilient Sampling*' addresses catastrophic forgetting by generating synthetic data through decoders of earlier-trained reconstructors. Within the VAE framework, this synthetic data, being both similar in nature but rich in diversity, is used alongside current task nodes for training. While the generated data might not inherently represent geographical graph structures, our graph learning technique ensures seamless integration. 3) '*Reconstruction-Based Replay*' employs a reconstructor to detect sensors that exhibit patterns not familiar to any expert. These nodes, determined by their reconstruction probability spanning all experts, are merged with the current task nodes, creating a dataset that captures patterns previously elusive to our expert models.

Through extensive experiments on a real-world long-term streaming network dataset, PEMSD3-Stream, we highlight the advantages of TFMoE. Our results demonstrate that TFMoE outperforms existing models, demonstrating significantly better performance and resilience against catastrophic forgetting. These findings validate the superiority of our proposed approach in addressing the challenges associated with continual learning in long-term streaming network scenarios, providing a robust and effective solution for traffic forecasting. The source code of TFMoE can be found `https://anonymous.4open.science/r/None3-28FA`.

## 2 RELATED WORK

**Traffic Flow Forecasting.** In traffic forecasting, traditional methods like ARIMA and SVR are popular but often miss intricate spatio-temporal patterns in road networks due to their reliance on historical data. Deep learning, specifically the combination of RNNs and CNNs as in (Zhang et al., 2018b; Yao et al., 2018a;b), has emerged to address these complexities. However, CNNs, being optimized for grid data, are not ideal for road networks. Thus, the focus has shifted to GCNs(Kipf & Welling, 2016; Defferrard et al., 2016; Bruna et al., 2013; Veličković et al., 2017) for capturing spatial relationships. For instance, DCRNN (Li et al., 2017) leverages a diffusion process on graphs for spatial dependencies and a sequence-to-sequence model for time. Similarly, STSGCN (Song et al., 2020) uses localized spatio-temporal subgraphs. Despite progress, most research remains on static traffic networks, overlooking the evolving nature of real-world traffic.

**Continual Learning.** Continual learning, or lifelong learning, targets systems that adapt to changing environments and accumulate knowledge, aiming primarily to prevent catastrophic forgetting—losing old knowledge while learning new. This domain has three main strategies: regularization-based, rehearsal-based, and architecture-based. Regularization methods (Kirkpatrick et al., 2017; Zenke et al., 2017) add terms to the loss function, limiting model parameter changes. Rehearsal methods (Robins, 1995; Rebuffi et al., 2017; Shin et al., 2017; Wu et al., 2018) utilize replay buffers or generative models to retain past data or tasks. Architecture strategies (Rusu et al., 2016; Mallya & Lazebnik, 2018) dynamically alter the model structure for new tasks. While continual learning has been recently explored in the graph domain (Zhou & Cao, 2021; Wang et al., 2020; 2022), a majority of studies focus on classification rather than regression, making direct application to streaming traffic networks challenging (Chen et al., 2021a; Wang et al., 2023a;b).

## 3 PROBLEM DEFINITION

In the context of long-term streaming traffic networks, we define $\tau \in (1, 2, \ldots, \mathcal{T})$ as an extended time interval, or a 'Task,' where the traffic network remains unchanged. These dynamic networks are sequenced as $G = \left(G^1, G^2, \ldots, G^{\mathcal{T}}\right)$, with each $G^\tau$ evolving from its predecessor $G^{\tau-1}$ via $G^\tau = G^{\tau-1} + \Delta G^\tau$. For a task $\tau$, its road network is defined as $G^\tau = (V^\tau, A^\tau)$, where $V^\tau$ is the set with $N^\tau$ traffic sensors and $A^\tau \in \mathbb{R}^{N^\tau \times N^\tau}$ is its adjacency matrix. Network variations are captured by $\Delta G^\tau = (\Delta V^\tau, \Delta A^\tau)$, with $\Delta V^\tau$ denoting new nodes. While the graph structure $G$ can rely on geographical-distance-based graphs, learning (or inferring) the structure from data is better suited for continual learning, as detailed in Section 4.2.1.

For a specific task denoted as $\tau$, the observed traffic flow data across the node set $N^\tau$ during the time span $(t - T' + 1) : t$ is represented by $\mathbf{X}_t^\tau = \{x_{1,t}^\tau, x_{2,t}^\tau, \ldots, x_{N^\tau,t}^\tau\} \in \mathbb{R}^{N^\tau \times T'}$. In this context, each element $x_{i,t}^\tau \in \mathbb{R}^{T'}$ represents the data for the $i^{th}$ node, spanning the preceding $T'$ time steps starting from time $t$. Likewise, data covering the subsequent time interval $(t+1) : (t+T)$ is represented as $\mathbf{Y}_t^\tau = \{y_{1,t}^\tau, y_{2,t}^\tau, \ldots, y_{N^\tau,t}^\tau\} \in \mathbb{R}^{N^\tau \times T}$, where each $y_{i,t}^\tau \in \mathbb{R}^T$ represents the upcoming $T$ time steps from $t+1$ for the $i^{th}$ node. Additionally, we introduce $\mathbf{X}_w^\tau = \{x_{1,w}^\tau, x_{2,w}^\tau, \ldots, x_{N^\tau,w}^\tau\} \in \mathbb{R}^{N^\tau \times week}$, where *week* refers to the total time steps obtained by segmenting an entire week into time intervals. Consequently, each $x_{i,w}^\tau \in \mathbb{R}^{week}$ denotes the initial one-week traffic data for the $i^{th}$ node. We select the first full week of data from Monday to Sunday in the training dataset.

Our primary objective is to develop a probabilistic regression model, parameterized by $\theta$, that can predict $\mathbf{Y}_t^\tau$ using both $\mathbf{X}_t^\tau$ and $\mathbf{X}_w^\tau$. This can be formally represented as $p(\mathcal{D}^\tau \mid \theta) = \prod_t \prod_{i=1}^{N^\tau} p\left(y_{i,t}^\tau \mid x_{i,t}^\tau, x_{i,w}^\tau; \theta\right)$, where $\mathcal{D}^\tau = \{(\mathbf{Y}_t^\tau, \mathbf{X}_t^\tau, \mathbf{X}_w^\tau)\}_t$ denotes the dataset corresponding to task $\tau$. Based on the above notations, we now describe our main approach—the Mixture of Experts (Jacobs et al., 1991) (MoE) framework. For a system with $K$ experts, the probability distribution $p(y_t^\tau \mid x_t^\tau, x_w^\tau; \theta)$, based on a gating mechanism, can be described as follows:

$$p(y_t^\tau \mid x_t^\tau, x_w^\tau; \theta) = \sum_{k=1}^{K} \underbrace{p(y_t^\tau \mid x_t^\tau, \eta = k; \theta)}_{\text{predictor}} \underbrace{p(\eta = k \mid x_w^\tau; \theta)}_{\text{gating term}}, \quad (1)$$

where $\eta$ denotes the expert indicator, and $p(\eta = k \mid x_w^\tau; \theta)$ denotes the probability assigned by the gating mechanism to expert $k$ for the given input $x_w^\tau$. The central challenge emerges when modeling the gating term $p(\eta = k \mid x_w^\tau; \theta)$. Rather than employing a single classifier model for modeling the gating term, our approach, inspired by (Lee et al., 2020), adopt the generative modeling: $p(y_t^\tau \mid x_t^\tau, x_w^\tau; \theta) = \sum_{k=1}^{K} p(y_t^\tau \mid x_t^\tau; \psi_k) \frac{p(x_w^\tau; \phi_k) p(\eta=k)}{\sum_{k'} p(x_w^\tau; \phi_{k'}) p(\eta=k')}$. Here, the function $p(x_w^\tau; \phi_k)$

represents the generative model, which is a reconstructor in our context. For clarity, we split the parameter $\theta$ into task-specific parameters, i.e., $\theta = \cup_{k=1}^{K}\theta_k$, where $\theta_k = \{\psi_k, \phi_k\}$. Additionally, we assume that for $p(\eta = k)$, the probability remains uniform. In the setting of long-term streaming traffic networks, the objective is to learn the sequence $(\theta^1, \theta^2, \ldots, \theta^{\mathcal{T}})$. Given the context of our continual learning scenario, it is important to note that during the learning of task $\tau$, we use parameters initialized with $\theta^{\tau-1}$ and conduct the learning process based on $\Delta N^{\tau} = |\Delta V^{\tau}|$.

## 4 PROPOSED METHOD: TFMoE

### 4.1 PRE-TRAINING STAGE

#### 4.1.1 RECONSTRUCTION-BASED CLUSTERING

In the initial phase of our training pipeline, we aim to group the $N^1 = |V^1|$ sensors from the first task (i.e., $\tau = 1$) into $K$ homogeneous clusters. We apply Deep Embedded Clustering (Xie et al., 2016), which enables dual learning of features and cluster assignments via deep networks. First, we pre-train an autoencoder feature extractor using a week's traffic data from each sensor. The reconstructed output for sensor $i$ is $\hat{x}_{i,w}^1 = R_{pre}(x_{i,w}^1)$, where $R_{pre}$ is the autoencoder and $x_{i,w}^1$ is the week-long data from sensor $i$. For optimization, we employ the MAE loss $\mathcal{L}_{recon} = \frac{1}{N^1}\sum_{i=1}^{N^1}\left\|x_{i,w}^1 - \hat{x}_{i,w}^1\right\|_1$. After training the feature extractor $R_{pre}$, we encode the week-long traffic data from sensor $i$, $x_{i,w}^1$, to a latent representation $\kappa_i = R_{pre;enc}(x_{i,w}^1)$, where $\kappa_i \in \mathbb{R}^{d_{\mathcal{Z}}}$.

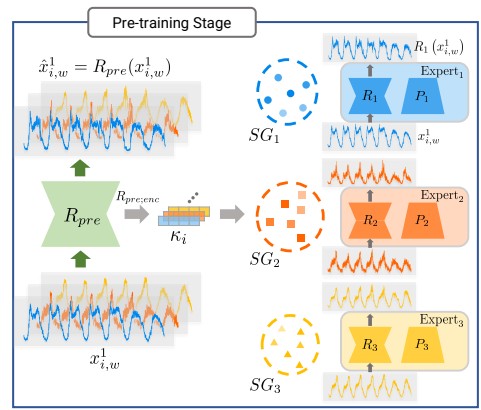

Figure 2: Architecture of Pre-training Stage.

$R_{pre;enc}$ is the encoder part of autoencoder $R_{pre}$. Using these representations, we perform balanced k-means clustering to get $K$ cluster centroids $[\mu_1; \ldots; \mu_K] \in \mathbb{R}^{K \times d_{\mathcal{Z}}}$, which serve as initial learnable parameters. We measure the soft cluster assignment probability between $\kappa_i$ and centroid $\mu_k$ using the Student's t-distribution (Van der Maaten & Hinton, 2008): $q_{ik} = \frac{\left(1+\|\kappa_i-\mu_k\|^2\right)^{-1}}{\sum_{k'}\left(1+\|\kappa_i-\mu_{k'}\|^2\right)^{-1}}$, where $q_{ik}$ denotes the probability of assigning sensor $i$ to cluster $k$. We further refine clustering via an auxiliary target distribution $p_i$ as follows (Xie et al., 2016): $p_{ik} = \frac{q_{ik}^2/\sum_{i'}q_{i'k}}{\sum_{k'}\left(q_{ik'}^2/\sum_{i'}q_{i'k'}\right)}$. The distribution $p_{ik}$ is strategically designed to augment the homogeneity within clusters, while giving precedence to data points associated with high confidence levels. For the purpose of achieving high-confidence assignments, we define the KL divergence loss between $q_i$ and $p_i$ as $\mathcal{L}_{cluster} = D_{KL}(P\|Q) = \sum_i\sum_k p_{ik}\log\frac{p_{ik}}{q_{ik}}$. Optimizing both the reconstruction loss $\mathcal{L}_{recon}$ and the clustering loss $\mathcal{L}_{cluster}$ aids in extracting meaningful patterns from data and clustering sensors with similar characteristics. The overall loss is: $\mathcal{L}_p = \mathcal{L}_{recon} + \alpha\mathcal{L}_{cluster}$.

#### 4.1.2 CONSTRUCTING EXPERTS

Utilizing the cluster assignment probability $q_i$, we established a hard assignment $c_i = \operatorname{argmax}_k(q_{ik})$ for each sensor $i$. Then, the group of sensors assigned to the $k$-th cluster is defined as $SG_k = \{i \mid c_i = k\}$, and sensors that belong to the same cluster shares homogeneous semantics. Accordingly, we assign an expert, i.e., $\text{Expert}_k = (R_k, P_k)$ comprising of a reconstructor $R_k$ and a predictor $P_k$, to each cluster $k$.

#### 4.1.3 TRAINING RECONSTRUCTOR OF EXPERT

Under the continual learning framework, we aim to train the reconstructor with two objectives:

**Sensor-Expert Matching.** In the ever-evolving traffic network landscape, the integration of a new sensor mandates the identification of an expert that is semantically compatible. To this end, we train the reconstructor $R_k$ to proficiently reconstruct the feature representations of its designated sensor group, denoted as $SG_k$. This strategic training ensures that, upon the introduction of new sensors, we can seamlessly identify the most appropriate expert that aligns with its semantic content.

**Forgetting-Resilient Sampling.** One of the primary concerns as we transition to subsequent tasks is to minimize catastrophic forgetting. Instead of directly storing the current data in memory, we

utilize a latent variable $z_k$ for each $\text{Expert}_k$ to generate data via $p\left(x_w \mid z_k; \phi_k\right)$. However, as the posterior distribution $p\left(z_k \mid x_w; \phi_k\right)$ is intractable, an approximation through the variational posterior $q\left(z_k \mid x_w; \xi_k\right)$, which is parameterized by $\xi_k$, becomes essential.

To address the aforementioned objectives, we model each expert's reconstructor as a Variational Autoencoder (VAE) (Kingma & Welling, 2013). For the $k$-th expert, the marginal likelihood for all data in $SG_k$ is given by $\log p\left(x_{(SG_k[1],w)}, x_{(SG_k[2],w)}, \ldots, x_{(SG_k[|SG_k|],w)}; \phi_k\right) = \sum_{i \in SG_k} \log p\left(x_{i,w}; \phi_k\right)$. For each data point $i$, it unfolds as:

$$\log p\left(x_{i,w}; \phi_k\right) = D_{KL}\left(q\left(z_k \mid x_{i,w}; \xi_k\right) \| p\left(z_k \mid x_{i,w}; \phi_k\right)\right)$$
$$+ \mathbb{E}_{q\left(z_k \mid x_{i,w}; \xi_k\right)}\left[-\log q\left(z_k \mid x_{i,w}; \xi_k\right) + \log p(x_{i,w}, z_k; \phi_k)\right] \qquad (2)$$

The Evidence Lower Bound (ELBO) for expert $k$ is then given by: $\mathcal{L}^k_{ELBO}\left(\phi_k, \xi_k\right) = \sum_{i \in SG_k} \mathbb{E}_{q(z_k \mid x_{i,w}; \xi_k)}\left[\log \frac{p(x_{i,w} \mid z_k; \phi_k)p(z_k; \phi_k)}{q(z_k \mid x_{i,w}; \xi_k)}\right]$. Training follows a similar approach to conventional VAEs, where the objective is to maximize $\mathcal{L}^k_{ELBO}$ for each cluster. Consequently, the overall objective for all clusters becomes $\mathcal{L}^{SG}_{ELBO} = \sum_{k=1}^K \mathcal{L}^k_{ELBO}\left(\phi_k, \xi_k\right)$. We set the prior using learnable parameters $\mu_k$ and $\Sigma_k$ as $p(z_k; \phi_k) = \mathcal{N}\left(z_k \mid \mu_k, \Sigma_k\right)$ to enhance the expressiveness of the latent space for the data assigned to each expert, where $\Sigma_k$ is a diagonal matrix. The method for sampling is described in detail in Section 4.2.3.

## 4.2 LOCALIZED ADAPTATION STAGE

### 4.2.1 TRAINING PREDICTOR OF EXPERT

We use the Mixture of Experts (Jacobs et al., 1991) framework to train each expert's predictor. This predictor takes the past $T'$ time steps of data $x^{\tau}_{i,t}$ from the $i^{th}$ sensor to forecast the next $T$ time steps $y^{\tau}_{i,t}$. We employ Graph Neural Network (GNN) layers to capture spatial sensor dependencies and 1D-Convolutional layers for the temporal dynamics of traffic. However, we argue that using predefined geographical-distance-based graphs for GNNs presents the following two issues:

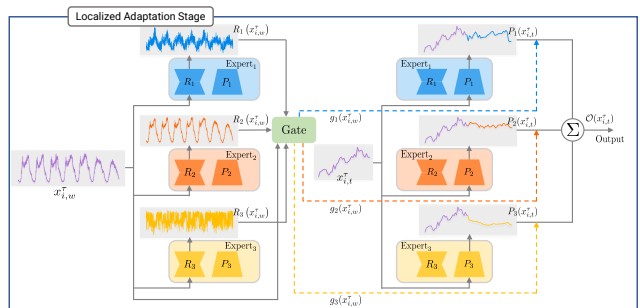

Figure 3: Architecture of Localized Adaptation Stage.

1. **Handling Newly Added Nodes.** When learning from newly added nodes within predefined graph structures rooted in actual geographical distances, Graph Neural Networks (GNNs) require the formation of *'subgraphs'* centered around these newly added nodes. Consequently, these newly added nodes inevitably establish many connections with pre-existing nodes from previous tasks. For this reason, when learning the current task using a GNN, it needs to access many pre-existing nodes. Although accessing as much data of pre-existing nodes as possible is indeed beneficial, it violates the goal of continual learning, whose main goal is to achieve optimal performance with minimal access to previous tasks.

2. **Lack of Graph Structure in Sampled Data.** Recall that to consolidate prior knowledge, we will sample data from a VAE decoder, $R_k$, previously trained on an earlier task (i.e., Forgetting-Resilient Sampling, which will be described in Section 4.2.3). However, as they are synthetically generated, they inherently lack the graph structural information, implying that the generated nodes would be isolated in a predefined geographical-distance-based graph. As a result, GNNs are hindered from maximizing their inherent strength of information propagation across nodes and edges.

**Solution: Graph Structure Learning.** We address these issues by adopting the graph structure learning mechanism (Zhu et al., 2021), leveraging the Gumbel softmax trick (Maddison et al., 2016; Jang et al., 2016), instead of using predefined geographical-distance-based graphs. For a given node $i$, its hidden embedding can be represented as: $e_i = L_e(x^{\tau}_{i,t})$, where $L_e(\cdot)$ is a linear transformation. Subsequently, the weight connecting nodes $i$ and $j$ is modeled as: $w_{ij} = L_w([e_i; e_j]) - \ln(-\ln(U))$, where $L_w(\cdot)$ is also a linear transformation. In this equation, $U$ is drawn from a uniform distribution, $U \sim \text{Uniform}(0, 1)$, which serves to introduce Gumbel noise into the model. In essence, the

adjacency matrix of the graph that links nodes $i$ and $j$ can be represented as $A_{ij} = \frac{\exp(w_{ij})}{\sum_{j'} \exp(w_{ij'})}$. The learned adjacency matrix inherently exhibits non-symmetry. To address this characteristic effectively, we adopt the Diffusion Convolution Layer (Li et al., 2017); a Graph Neural Network (GNN) layer designed to model spatial dependencies via a diffusion process. Given an input graph signal $\mathcal{X} \in \mathbb{R}^{N \times D}$ and an adjacency matrix $A_{ij}$, the diffusion convolution operation can be defined as: $\mathcal{X}_{:,p} \star_A f_\zeta = \sum_{m=0}^{M-1} \left( \zeta_{m,1} \left( \boldsymbol{D}_O^{-1} A \right)^m + \zeta_{m,2} \left( \boldsymbol{D}_I^{-1} A^\top \right)^m \right) \mathcal{X}_{:,p}$ for $p \in \{1, \cdots, D\}$, where $\zeta \in \mathbb{R}^{M \times 2}$ denotes a trainable matrix, and $M$ signifies the number of diffusion steps. The matrices $\boldsymbol{D}_O^{-1} A$ and $\boldsymbol{D}_I^{-1} A^\top$ function as state transition matrices. Specifically, $\boldsymbol{D}_O = \mathrm{diag}(A\mathbf{1})$ and $\boldsymbol{D}_I = \mathrm{diag}(A^\top \mathbf{1})$ are the out-degree and in-degree diagonal matrices, respectively. Here, $\mathbf{1} \in \mathbb{R}^N$ is a column vector with all elements set to one. Building upon the convolution operation, the Diffusion Convolutional Layer is defined as: $\mathcal{H}_{:,q} = \sum_{p=1}^{D} \mathcal{X}_{:,p} \star_A f_{\Lambda_{q,p,:,:}}$ for $q \in \{1, \cdots, D'\}$, where $\mathcal{H} \in \mathbb{R}^{N \times D'}$ represents our desired output and $\Lambda \in \mathbb{R}^{D' \times D \times M \times 2} = [\zeta]_{q,p}$ denotes the parameter tensor.

**Predictor:** For predictor, we utilized a single Diffusion Convolutional Layer for capturing spatial dependencies among sensors. This was followed by two 1D-Conv layers to capture traffic's temporal dynamics. Though our predictor is simplified to validate the effectiveness of our proposed framework, more advanced prediction models can be integrated. Within each expert, $\mathrm{Expert}_k = (R_k, P_k)$, the predictor $P_k$ takes the past $T'$ time steps of data $x_{i,t}^\tau$ to produce its output $P_k\left(x_{i,t}^\tau\right)$.

**Reconstruction-based Gating:** Having defined the predictors $\{P_1, P_2, ..., P_K\}$, we now describe the reconstruction-based gating mechanism that leverages the reconstructors $\{R_1, R_2, ..., R_K\}$, trained in the previous section, to assign weights to the the predictions of each predictor. The gating weights assigned to each predictor are defined as follows: $g_k(x_{i,w}^\tau) = \frac{p\left(x_{i,w}^\tau; \phi_k\right)}{\sum_{k'} p\left(x_{i,w}^\tau; \phi_{k'}\right)}$, Here, $x_{i,w}^\tau$ denotes the one-week traffic data of the sensor $i$ in task $\tau$ from the first Monday to Sunday. Moreover, $g_k(x_{i,w}^\tau)$ represents the weight given to sensor $i$ in task $\tau$ for the predictor $P_k$. A large value of $g_k(x_{i,w}^\tau)$ implies that the prediction from $P_k$ is particularly important for the final prediction of sensor $i$. Utilizing these gating weights, the final prediction, which integrates the outputs from all predictors $\{P_1, P_2, ..., P_K\}$, is defined as follows: $\mathcal{O}\left(x_{i,t}^\tau\right) = \sum_{k=1}^{K} g_k\left(x_{i,w}^\tau\right) P_k\left(x_{i,t}^\tau\right)$. In other words, the final prediction on $x_{i,t}^\tau$ is generated by the weighted sum of the predictions made by the predictors $P_k$, and the weight is determined by how well the first week traffic data of the sensor $i$ is reconstructed by the reconstructor $R_k$, which is denoted by $g_k(x_{i,w}^\tau)$. Correspondingly, the loss function employed to train the entire set of experts can be formulated as follows: $\mathcal{L}_{\mathcal{O}} = \sum_{i=1}^{N^\tau} \left\| \mathcal{O}\left(x_{i,t}^\tau\right) - y_{i,t}^\tau \right\|_1$.

**Discussion: Learning on the Expanding Traffic Network.** Real-world traffic networks expand as new sensors emerge in surrounding areas. While these sensors can reflect known traffic patterns, they often introduce new dynamics. In the first task (i.e., $\tau = 1$), the entire graph $G^1 = \left(V^1, A^1\right)$ is known. But re-training the network for each subsequent task is impractical. Our goal is to efficiently retain prior knowledge while accommodating new patterns via continual learning.

### 4.2.2 RECONSTRUCTION-BASED KNOWLEDGE CONSOLIDATION

Inspired by the Learning without Forgetting (LwF) approach (Li & Hoiem, 2017), we propose a novel strategy termed the "reconstruction-based consolidation loss" to retain previously acquired knowledge while adapting to new tasks. We use $LG_k^\tau$ to denote the localized group associated with the $k$-th reconstructor for the training of task $\tau$, i.e., $LG_k^\tau = \left\{ i \,\middle|\, \underset{j}{\arg\max}\, p\left(x_{i,w}^\tau; \phi_j^{(\tau-1)}\right) = k, i \in \Delta V^\tau \right\}$, where $\phi_j^{(\tau-1)}$ indicates the parameters of the $j$-th reconstructor that have been optimized upon completion of training up to the $(\tau - 1)$-th task. A localized group collects the newly added nodes $\Delta V^\tau$ in the current task based on the reconstruction probability determined by the reconstructor that was trained and optimized in the previous task $(\tau - 1)$. The reconstruction-based consolidation loss using VAE, which incorporates the use of the localized group, is developed in a manner similar to Section 4.1.3. For the $k$-th expert, the marginal likelihood for all data in $LG_k$ is represented as: $\log p\left(x_{(LG_k[1],w)}, \ldots, x_{(LG_k[|LG_k|],w)}; \phi_k\right) = \sum_{i \in LG_k} \log p\left(x_{i,w}; \phi_k\right)$. The Evidence Lower Bound (ELBO) for all experts using the variational distribution $q$ is as: $\mathcal{L}_{ELBO}^{LG} = \sum_{k=1}^{K} \sum_{i \in LG_k} \mathbb{E}_{q(z_k|x_{i,w}; \xi_k)} \left[ \log \frac{p(x_{i,w}|z_k; \phi_k) p(z_k; \phi_k)}{q(z_k|x_{i,w}; \xi_k)} \right]$.

The role of the consolidation loss is as follows: The reconstructor classifies newly added nodes $\Delta V^{\tau}$ of the current task into localized groups $LG_k^{\tau}$, based on the optimized parameters $\phi_k^{(\tau-1)}$ from the previous task. Then, during the learning process of reconstructor parameters $\phi_k^{\tau}$ in the current task $\tau$, consolidation loss strives to maintain the nodes belonging to these localized groups as much as possible. In essence, this helps preserve knowledge from the previous task while learning new information in the current task. The optimization objective for training the entire model is given by the loss function $\mathcal{L} = \mathcal{L}_{\mathcal{O}} - \beta \mathcal{L}_{ELBO}^{LG}$. Here, where $\beta$ is the weight of the consolidation loss. For the first task, since training is conducted on nodes partitioned into $SG_k$, the final loss is represented as $\mathcal{L} = \mathcal{L}_{\mathcal{O}} - \beta \mathcal{L}_{ELBO}^{SG}$.

### 4.2.3 FORGETTING-RESILIENT SAMPLING

Next, we introduce 'forgetting-resilient sampling,' which is an additional methodology that mitigates catastrophic forgetting. The core idea is to utilize the decoders of reconstructors that were trained on earlier tasks to generate synthetic data samples. Within the context of our VAE-based reconstructor, for each $\text{Expert}_k$, we can sample $n_s/k$ instances of the latent variable $\{z_{k,1}, z_{k,2}, \ldots, z_{k,n_s/k}\} \sim p\left(z_k; \phi_k^{(\tau-1)}\right)$. From each $z_{k,i}$, data samples are generated according to $x_{w_{k,i}} \sim p\left(x_w \mid z_{k,i}; \phi_k\right)$. Thus, the dataset sampled for $\text{Expert}_k$ can be defined as $X_k^s = \left\{x_{w_{k,1}}, x_{w_{k,2}}, \ldots, x_{w_{k,n_s/k}}\right\}$. By aggregating over all experts, the entire sampled dataset is given by $X^s = \bigcup_{k=1}^{K} X_k^s$, where $|X^s| = n_s$. Note that we set $n_s$ as a hyperparameter in our model. As we train our model, we integrate this generated data with the new nodes $\Delta V^{\tau}$ from the current task $\tau$. Since our generated data encapsulates a week's worth of information, we have ensured synchronization of its temporal aspects when feeding it into the predictor, implementing appropriate data slicing for temporal alignment.

An important point to stress is that, being synthetic, our generated data do not inherently have a graph structure determined by the actual geographical distances. Yet, by adopting the graph learning methodology detailed in Section 4.2.1, we can adeptly address this limitation. Moreover, when updating embeddings using a GNN, it is essential to possess data that are both similar in nature and rich in diversity. The sampling approach facilitated by our VAE guarantees efficient production of such diverse samples for each expert, which is corroborated by our experiments in Section F.

### 4.2.4 RECONSTRUCTION-BASED REPLAY

In the context of continual learning, one characteristic of streaming traffic networks is that even pre-existing sensors display new patterns over long-term periods (e.g., several years) due to various factors such as urban development. While we strive to minimally access information from previous tasks, these sensors inherently offer indispensable information for our model training. Our main focus is on the pre-existing sensors that exhibit patterns distinct enough that they are universally unfamiliar to all experts, posing significant challenges in handling. To systematically identify these sensors, we deploy a reconstructor described as follows:

$$V_R = \{v[1], v[2], ..., v[n_r]\}, \text{ where}$$
$$\sum_k \log p\left(x_{v[1]}; \phi_k^{(\tau-1)}\right) \leq \sum_k \log p\left(x_{v[2]}; \phi_k^{(\tau-1)}\right) \leq \cdots \leq \sum_k \log p\left(x_{v[N^{(\tau-1)}]}; \phi_k^{(\tau-1)}\right) \quad (3)$$
$$\text{and } \forall j \in \{1, 2, ..., N^{(\tau-1)}\}; v[j] \in V^{(\tau-1)},$$

where $n_r$ is a hyperparameter dictating the number of elements in set $V_R$. The set $V_R$ comprises nodes, sorted in ascending order based on their reconstruction probability across all experts' reconstructors, and it is truncated to retain only the first $n_r$ entries. For training, nodes from $V_R$ are combined with the nodes of the current task, represented as $\Delta V^{\tau}$, along with the nodes sampled as discussed in Section 4.2.3. Consequently, the total number of nodes employed for training is given by $\Delta N^{\tau} + n_s + n_r$.

## 5 EXPERIMENTS

**Dataset**. Please refer to Appendix A for details regarding dataset.

**Baselines.** Please refer to Appendix B for details regarding baselines.

**Evaluation Protocol.** Please refer to Appendix C for details regarding evaluation protocol.

**Implementation Details.** Please refer to Appendix D for details regarding implementation details.

Table 1: Performance of traffic flow forecasting on PEMSD3-Stream dataset. The model names are appended with their corresponding pre-existing sensor access ratios, e.g., TFMoE (1%) indicates the TFMoE model with a 1% access ratio. Importantly, $\gamma$ represents the ratio of pre-existing sensors accessed due to the subgraph structure. Please note that, in line with standard practice in traffic forecasting research, we use the exact prediction values corresponding to their exact time frames: 15-minute, 30-minute, and 60-minute, for their respective evaluation metrics[1].

| PEMSD3-Stream | | 15 min | | | 30 min | | | 60 min | | | Training Time (sec) |
|---|---|---|---|---|---|---|---|---|---|---|---|---|
| Model | | MAE | RMSE | MAPE | MAE | RMSE | MAPE | MAE | RMSE | MAPE | | |
| Retrained | GRU | 13.51 | 21.83 | 18.19 | 15.80 | 25.89 | 21.05 | 23.47 | 35.53 | 28.50 | | 1824 |
| | DCRNN | 12.38 | 18.98 | 16.64 | 13.78 | 21.20 | 18.26 | 16.88 | 26.76 | 21.13 | | 50K+ |
| | STSGCN | 12.95 | 19.84 | 17.20 | 13.53 | 21.05 | 18.15 | 16.05 | 25.89 | 21.10 | | 50K+ |
| | Retrained-TFMoE | 12.37 | 20.32 | 16.38 | 13.80 | 22.75 | 17.95 | 17.02 | 27.91 | 21.88 | | 846 |
| Online | Static-TFMoE | 14.43 | 23.50 | 21.22 | 16.19 | 26.72 | 23.05 | 20.78 | 34.29 | 29.46 | | 209 |
| | Expansible-TFMoE | 14.68 | 23.84 | 21.38 | 16.74 | 27.43 | 23.93 | 21.77 | 35.69 | 31.08 | | 239 |
| | TrafficStream (10+$\gamma$%) | 13.94 | 22.69 | 20.02 | 15.91 | 26.03 | 22.52 | 21.12 | 34.04 | 30.16 | | 206 |
| | TFMoE (1%) | **12.58** | **20.63** | **16.63** | **14.12** | **23.27** | **18.40** | **17.65** | **28.95** | **22.62** | | **255** |

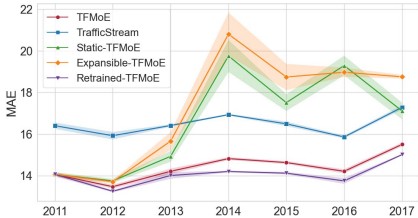

Figure 4: The MAE of traffic flow forecasting over consecutive years.

| Model | MAE | RMSE | MAPE | Use Subgraphs? | Store Features? |
|---|---|---|---|---|---|
| Retrained-TFMoE | 14.06 | 23.29 | 18.30 | ✗ | ✗ |
| Static-TFMoE | 16.63 | 27.68 | 23.70 | ✗ | ✗ |
| Expansible-TFMoE | 17.24 | 28.56 | 24.76 | ✗ | ✗ |
| TrafficStream (10+$\gamma$%) | 16.35 | 26.60 | 24.23 | ✓ | ✓ |
| PECMP (10+$\gamma$%) | 16.02* | 26.51* | 22.30* | ✓ | ✓ |
| TFMoE(1%) | **14.42** | **23.90** | **18.78** | ✗ | ✗ |

Table 2: 60-minute average prediction performance using the TrafficStream predictor structure across various models. * indicates values reported from the PECMP paper.

## 5.1 EXPERIMENTAL RESULTS

Table 1 demonstrates the forecasting capabilities of each model at different time horizons (15-, 30-, and 60-minutes ahead), captured by average MAE, RMSE, and MAPE over 7 years. Figure 4 visualizes these metrics from 2011 to 2017. The 60-minute average outcomes using the TrafficStream predictor structure across various models are detailed in Table 2. As the official source code of PECMP is not available, we include only comparable reported values.

We have the following observations: **1)** Static-TFMoE underperforms because it uses only 2011 data for predictions up to 2017. This highlights the necessity of integrating new data for accurate forecasting. **2)** Results from Expansible-TFMoE reveal that depending solely on newly added sensor data each year, without referencing historical knowledge, degrades performance due to catastrophic forgetting. **3)** Existing models (i.e., TrafficStream and PECMP) connect newly added nodes with pre-existing nodes by constructing subgraphs around them, while TFMoE does not require an access to any pre-existing nodes thanks to the graph structure learning. Besides, existing models also construct subgraphs around replayed nodes, which further adds the number of accessed pre-existing nodes (denoted by $\gamma^2$), while TFMoE merely accesses the replayed nodes without construction of subgraphs. Note that existing models use 10% of the number of pre-existing sensors for replay, while TFMoE only uses 1%. In short, TFMoE outperforms existing models even with a significantly limited access to pre-existing nodes. **4)** To detect and replay sensors with patterns that are either similar or different between the current and the previous year, existing models typically require storing features from all sensors of the previous year in a separate memory. However, as elaborated in Section 4.2.4, our approach uses a VAE-based reconstructor, allowing us to analyze and compare with past patterns using only the data from the current task. This eliminates the need for dedicated memory storage for historical data. From the perspective of continual learning, which aims to minimize memory usage for past tasks, this presents a significant advantage. **5)** While other

---

[1]In previous studies, authors report the average metrics considering the interval of 5-minute. For instance, in the case of the 15-minute, they computed an average of predictions of 5-minute, 10-minute and 15-minute. In this work, we follow the standard practice in traffic forecasting research, for example, we compute the metric based on only the predictions made at 15-minute. This difference in calculation method may account for any discrepancies in numeric values when compared to the results of previous studies.

[2]Our empirical observations indicate that for models employing subgraphs, the average access rate to pre-existing sensors exceeds 20%, implying $\gamma\% > 20\%$. This is because replayed nodes also possess subgraphs.

Table 3: Component analysis of TFMoE.

| Method | 15 min | | | 30 min | | | 60 min | | | Training Time (sec) |
|---|---|---|---|---|---|---|---|---|---|---|
| | MAE | RMSE | MAPE | MAE | RMSE | MAPE | MAE | RMSE | MAPE | |
| w/o Consol | 13.60 | 22.02 | 18.44 | 15.31 | 24.88 | 20.46 | 19.52 | 31.52 | 25.70 | 224 |
| w/o Sampling | 12.90 | 21.07 | 17.04 | 14.55 | 23.89 | 18.96 | 18.43 | 30.03 | 23.60 | 245 |
| w/o Replay | 14.90 | 24.17 | 22.33 | 16.99 | 27.77 | 25.25 | 22.11 | 36.16 | 33.04 | 249 |
| TFMoE | 12.58 | 20.63 | 16.63 | 14.12 | 23.27 | 18.40 | 17.65 | 28.95 | 22.62 | 255 |

methods employ various strategies against catastrophic forgetting, TFMoE stands superior, accessing only 1% of the pre-existing nodes corresponding to past tasks. This signifies its robustness and suitability for real-world traffic forecasting.

## 5.2 MODEL ANALYSIS

**Component Analysis:** We delve deeply into the individual components of TFMoE. To systematically evaluate the contribution of each component to the overall performance, we introduce four model variations: (i) TFMoE which integrates all the components, (ii) w/o Consol which excludes the Consolidation Loss (detailed in Section 4.2.2), (iii) w/o Sampling which operates without the Sampling mechanism (as discussed in Section 4.2.3), and (iv) w/o Replay that does not employ the Replay strategy (outlined in Section 4.2.4). The comparative performances of these models are presented in Table 3. We observe that each component plays a pivotal role in enhancing the model's efficacy, as elucidated in the respective sections. Notably, TFMoE, with its full suite of components, stands out as the top-performing model, validating the efficacy of our proposed methodologies.

**Effect of Consolidation Loss on Expert Utilization:** In this section, we evaluate the impact of the consolidation loss described in Section 4.2.2 on expert utilization within TFMoE, emphasizing its role in addressing catastrophic forgetting. Our ablation study, depicted in Figure 5, employs two heatmaps. The $x$-axis spans the years 2011-2017, and the $y$-axis represents our four experts. Each cell in the heatmap indicates the number of sensors allocated to an expert during training, based on the lowest recon-

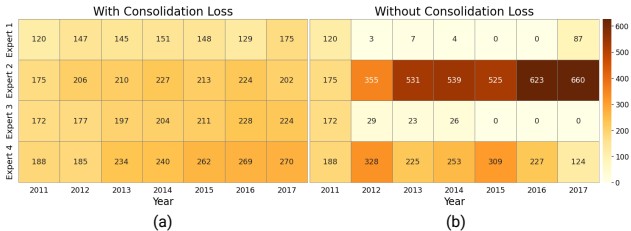

Figure 5: Comparison of sensor allocation to experts over the years (2011-2017) (a) With consolidation loss and (b) Without consolidation loss.

struction error. Figure 5 (a) and (b) contrast scenarios with and without the consolidation loss, respectively. We observe that incorporating the consolidation loss results in a balanced sensor distribution across experts throughout the years, which helps address catastrophic forgetting by maintaining each expert's unique contributions. Without the consolidation loss, sensors predominantly cluster around one expert after the first year, indicating a failure in diverse expert utilization and the onset of catastrophic forgetting. In essence, the consolidation loss is pivotal for balanced expert engagement in TFMoE, preventing dominance by any single expert and fostering effective continual learning.

## 6 CONCLUSION

In this paper, we introduced the Traffic Forecasting Mixture of Experts (TFMoE), a novel continual learning approach designed specifically for long-term streaming networks. Informed by real-world traffic patterns, TFMoE generates a specialized expert model for each homogeneous group, effectively adapting to evolving traffic networks and their inherent complexities. To overcome the significant obstacle of catastrophic forgetting in continual learning scenarios, we introduce three complementary mechanisms: 'Reconstruction-Based Knowledge Consolidation Loss', 'Forgetting-Resilient Sampling', and 'Reconstruction-Based Replay mechanisms', which allow TFMoE to retain essential prior knowledge effectively while seamlessly assimilating new information. The merit of our approach is validated through extensive experiments on a real-world long-term streaming network dataset, PEMSD3-Stream. Not only did TFMoE demonstrate superior performance in traffic flow forecasting, but it also showcased resilience against catastrophic forgetting, a key factor in the continual learning of traffic flow forecasting in long-term streaming networks. As such, our model offers a potent and efficient approach to tackling the evolving challenges of traffic flow forecasting.

**Ethics Statement** In alignment with ICLR's principles, our research endeavors to offer positive contributions to society. We uphold rigorous scientific standards, ensuring accurate, transparent, and reproducible outcomes. Every effort was made to minimize potential harm. Throughout our research, we emphasized fairness and meticulously adhered to privacy and confidentiality standards in data collection and usage.

**Reproducibility Statement** To ensure reproducibility, we detailed the datasets used in our experiments and the evaluation protocol in Appendices A and C , respectively. Further, the implementation details of our model are provided in Appendix D . The source code for our TFMoE is available at `https://anonymous.4open.science/r/None3-28FA`.

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

APPENDIX

## A    DATASET

In order to evaluate the performance of TFMoE, we conduct experiments on the PEMSD3-Stream dataset, which is a real-world highway traffic dataset collected by the Caltrans Performance Measurement System (PeMS) in real time every 30 seconds (Chen et al., 2001). The traffic data are aggregated into 5-minute intervals, resulting in 12 time steps per hour. PEMSD3-Stream dataset consists of traffic flow information in the North Central Area from 2011 to 2017. Consistent with previous studies, we select data from July 10th to August 9th annually. The input data is rescaled using Z-score normalization.

Table 4: Dataset statistics of pre-defined graph structure.

| Year | 2011 | 2012 | 2013 | 2014 | 2015 | 2016 | 2017 |
|---|---|---|---|---|---|---|---|
| # Nodes | 655 | 715 | 786 | 822 | 834 | 850 | 871 |
| # Edges | 3926 | 4390 | 4918 | 5218 | 5294 | 5404 | 5572 |

**Predefined Graph Structures.**

The PEMSD3-Stream traffic network is continually expanding, meaning that sensors installed in the $\tau$-th year continue to exist in the following years. Previous study utilized adjacency matrices grounded in geographical distances to represent the predefined graph structures. However, our analysis reveals that these matrices contain a high number of connected components. To reduce this number and create a more realistic network, we adopt a strategy where each individual sensor is connected to its $k$ nearest sensors based on geographical proximity, each connection (edge) having a weight of 1. In other words, if we denote the adjacency matrix in the $\tau$-th year as $A^\tau$, it is structured as follows: $A^\tau_{ij} = 1$ if $j \in$ Nearest$_k(i)$ else 0, where Nearest$_k(i)$ represents the set of $k$-nearest nodes of node $i$. Subsequently, to ensure the symmetry of the adjacency matrix, we carried out the operation $A = A \cup A^T$. We tune the value of $k$ by investigating the real-world connectivity patterns on the map, and we set $k$ to 5 for our experiments. Figure 6 provides a visual comparison between our predefined graph construction and the one from TrafficStream. For fair comparisons, when conducting experiments that utilize predefined graph structures, we employ our newly defined graph structure for all models to ensure accurate performance. A detailed summary of the datasets and the predefined graph structure can be found in Table 4.

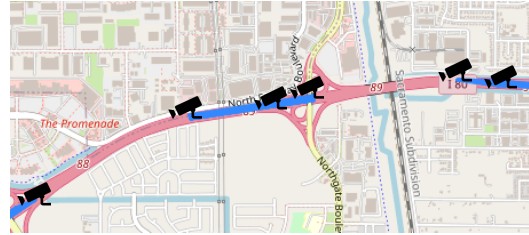

(a) Predefined graph structure provided by TrafficStream

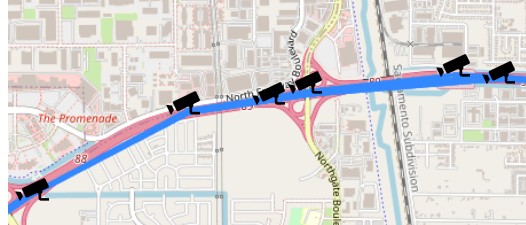

(b) Our modified predefined graph structure

Figure 6: Comparison between the predefined graph structure provided by TrafficStream and our modified predefined graph structure.

## B    BASELINES

We compare TFMoE with the following baselines:

- **GRU (Chung et al., 2014):** GRU operates by leveraging gated recurrent units, which enable it to effectively manage sequential data by adapting the information flow through its internal memory mechanism. For the training of this GRU, we utilize data from every node each year.

- **DCRNN (Li et al., 2017):** DCRNN integrates both spatial and temporal dependency by using diffusion convolution and encoder-decoder architecture. For the training of this DCRNN, we utilize data from every node each year.

- **STSGCN (Song et al., 2020):** STSGCN employs a localized spatial-temporal graph convolutional module, which allows the model to directly and concurrently capture intricate localized spatial-temporal correlations. For the training of this STSGCN, we utilize data from every node each year.

- **TrafficStream (Chen et al., 2021a):** The first work that applies continual learning techniques to a streaming traffic dataset. In order to mitigate catastrophic forgetting, TrafficStream proposes the use of Elastic Weight Consolidation (EWC) and a replay mechanism utilizing Jensen-Shannon divergence.

- **PECPM (Wang et al., 2023b):** PECPM utilizes a bank module and Elastic Weight Consolidation (EWC) for pattern storage in evolving traffic networks, allowing the model to adapt seamlessly with pattern expansion and consolidation.

- **Retrained-TFMoE:** We retrain TFMoE every year with all nodes given in each year. We initialize each year's model with the parameters learned from the previous year's model.

- **Static-TFMoE:** We train TFMoE solely on data from the first year (2011) and then directly use the trained model, without further training, to predict the traffic flow of all subsequent years.

- **Expansible-TFMoE:** We train TFMoE in an online manner each year, utilizing only data from the newly added sensors, while initializing each year's model with the parameters learned from the previous year's model. This model is equivalent to TFMoE without sampling and replay, as described in Sections 4.2.3 and 4.2.4, respectively.

## C  EVALUATION PROTOCOL

We leverage three standard performance metrics for model evaluation: mean absolute error (MAE), root mean squared error (RMSE), and mean absolute percentage error (MAPE). The datasets are divided into training, validation, and test sets with a distribution ratio of 6:2:2. Utilizing datasets aggregated into 5-minute intervals, our model uses one hour of historical data (12 time steps) to make predictions for the subsequent hour (12 time steps). Experiments are conducted on 13th Gen Intel(R) Core(TM) i9-13900K and NVIDIA GeForce RTX 4090. We conducted each experiment five times and report the mean performance.

## D  IMPLEMENTATION DETAILS

In our proposed framework, the number of clusters (i.e., Experts) is set to $K = 4$. In the 'Reconstruction-Based Clustering' (Section 4.1.1), an AutoEncoder structure is adopted for the pre-training reconstructor, where both the encoder and decoder are designed with three MLPs each. In the 'Training Reconstructor' (Section 4.1.3), the reconstructor is based on a Variational AutoEncoder (VAE) architecture, with both the encoder and decoder again made up of three MLPs. For both the AutoEncoder and VAE, the encoded latent hidden dimension is set to 32. The coefficient for the cluster loss, $\alpha$, is set to 0.0001. In the 'Training Predictor of Expert' (Section 4.2.1), the predictor comprises one GNN layer and two 1-D convolution layers, with a consistent hidden dimension of 32 throughout. The diffusion convolution operation's diffusion step is set to $M = 2$. The consolidation loss, $\beta$, for the 'Reconstruction-Based Knowledge Consolidation' (Section 4.2.2) is fixed to 10. In the 'Forgetting-Resilient Sampling' (Section 4.2.3), the number of samples, $n_s$, is set to 9% of the current task graph size, while in the 'Reconstruction-Based Replay' (Section 4.2.4), the number of replays, $n_r$, is set to 1% of the current task graph size.

Training was executed with 80 epochs for the first task and 10 epochs for the following tasks. We employed a batch size of 128 and utilized the Adam (Kingma & Ba, 2014) optimizer, setting learning rates of 0.001, 0.0001, and 0.01 for the pre-training reconstructor, reconstructor, and predictor, respectively.

## E  ANALYSIS OF RECONSTRUCTOR AND PREDICTOR OUTPUTS

We perform a detailed analysis of the performance of TFMoE with a specific emphasis on the individual outputs generated by the experts. We have the following observations in Figure 7: **1)** Upon close inspection of the one-week traffic flow data from sensor 126 (i.e. Figure 7 (a)), it becomes evident that expert 2 has managed to achieve the most accurate reconstruction. **2)** Moreover, a detailed

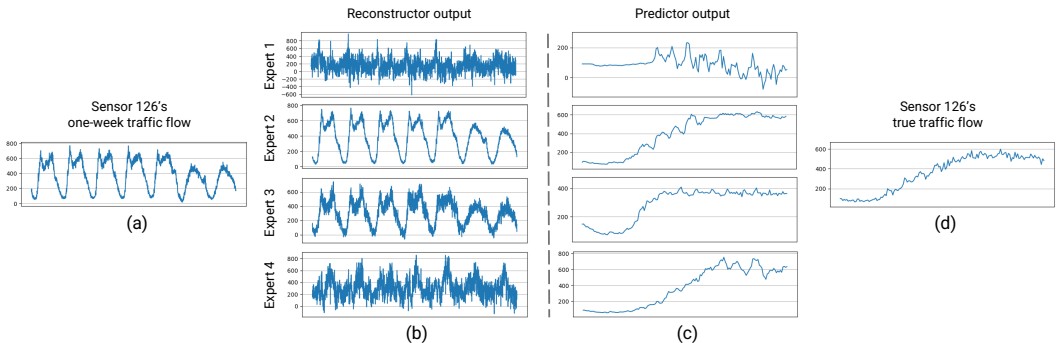

Figure 7: (a) Actual one-week traffic flow data measured from sensor 126 in the year 2012. (b) One-week traffic flow reconstructed by each expert's Reconstructor. (c) Traffic flow forecasted by each expert's Predictor. (d) Actual traffic flow data measured from sensor 126.

examination of the traffic flows predicted by each expert's Predictor (i.e., Figure 7 (c)) demonstrates that expert 2's predictor closely mirrors the true traffic flow (i.e., Figure 7 (d)). **3)** While the predicted traffic flow from predictor 2 and 3 may seem similar, a closer observation of the scale on the y-axis reveals that the output from expert 2's predictor aligns more precisely with the true traffic data. Through the above comprehensive analyses, we visually underscore the notion that the experts proficient in reconstruction also tend to provide more accurate predictions. This in-depth evaluation helps elucidate the functioning of the components of TFMoE and their individual contributions to the overall performance.

# F    VISUALIZATION OF SYNTHETIC DATA GENERATED THROUGH DECODERS

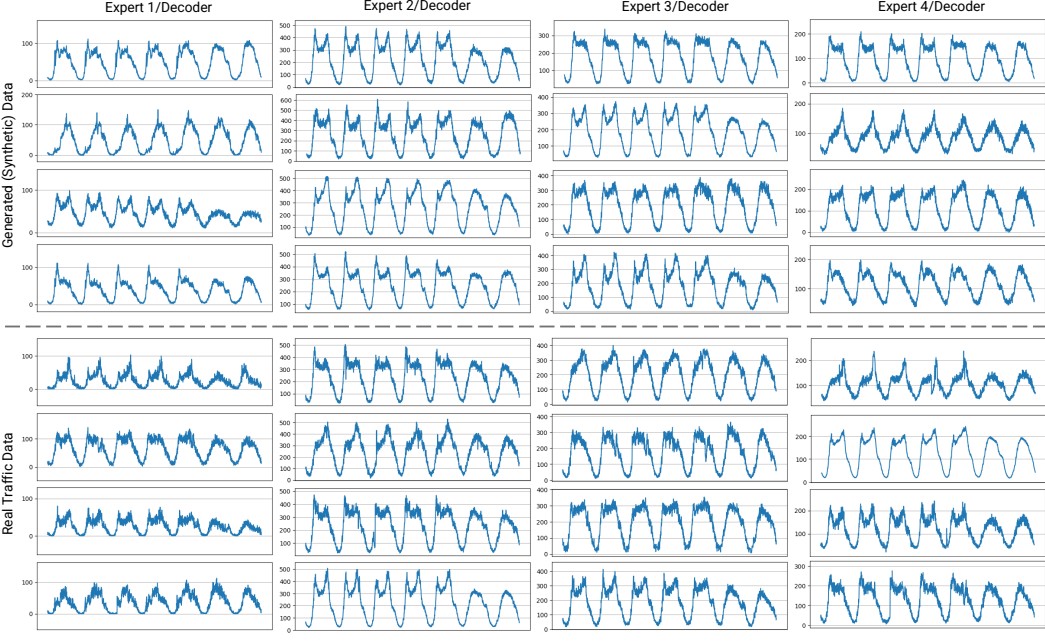

Figure 8: Visualization of synthetic data generated through decoders. Above the dashed line, we present plots of synthetic samples drawn using $x_w \sim p(x_w \mid z_k; \phi_k)$, where $z_k \sim p(z_k; \phi_k)$ for each Expert. Below the dashed line, the plots showcase the actual traffic data assigned to the respective Expert.

To assess how well the synthetic data mirrors the characteristics of real traffic data assigned to each Expert, we leveraged the decoders of individual Experts. Specifically, synthetic samples were drawn using $x_w \sim p\left(x_w \mid z_k; \phi_k\right)$, where $z_k \sim p\left(z_k; \phi_k\right)$. By examining Figure 8, we can observe that the synthesized data reflects the attributes of the real traffic data for each Expert. This is evident both from the y-axis scale and the overall shape of the graphs. Building on our discussions in Section 4.2.3, our objective is to generate data that is both similar in nature but rich in diversity. The synthetic data showcased in the figure aligns well with this objective, suggesting that our approach is adept at preserving past memories in a continual learning framework.

## G  OPTIMIZING EXPERT SELECTION

Table 5: Performance metrics (MAE, RMSE, MAPE) and training time over different numbers of experts.

| # of Expert | 15 min | | | 30 min | | | 60 min | | | Training Time (sec) |
|---|---|---|---|---|---|---|---|---|---|---|
| | MAE | RMSE | MAPE | MAE | RMSE | MAPE | MAE | RMSE | MAPE | |
| K = 1 | 13.16 | 21.67 | 17.29 | 15.02 | 24.93 | 19.59 | 19.58 | 32.28 | 24.97 | 76 |
| K = 2 | 12.83 | 21.07 | 16.82 | 14.54 | 24.02 | 18.84 | 18.44 | 30.24 | 23.67 | 140 |
| K = 3 | 12.73 | 20.93 | 16.79 | 14.32 | 23.65 | 18.73 | 18.01 | 29.54 | 23.14 | 209 |
| K = 4 | 12.58 | 20.63 | 16.63 | 14.12 | 23.27 | 18.40 | 17.65 | 28.95 | 22.62 | 255 |
| K = 5 | 12.49 | 20.51 | 16.57 | 14.02 | 23.16 | 18.28 | 17.41 | 28.59 | 22.46 | 338 |
| K = 6 | 12.46 | 20.42 | 16.48 | 13.94 | 22.97 | 18.18 | 17.25 | 28.32 | 22.19 | 391 |

In our exploration of the impact of expert count in TFMoE, a notable trend emerges. Table 5 reveals that as we increase the number of experts, prediction accuracy indicators, such as MAE, RMSE, and MAPE, generally decrease. This enhanced performance can be attributed to each expert's ability to specialize in distinct clusters of data, thus improving the overall accuracy. However, this increased granularity comes at a cost: heightened training times in direct proportion to the number of experts.

Interestingly, the benefits begin to plateau between the inclusion of the fourth and fifth expert. Despite the added computational cost, there is no significant difference in error metrics. Consequently, it appears that utilizing four experts strikes the ideal balance between accuracy and computational overhead. This observation underscores the importance of expert selection, balancing predictive accuracy against model complexity.

A potential reason behind the diminishing returns in performance, as the expert count grows, is redundancy. Ideally, we aim for experts to be distinctly trained to handle varying traffic patterns. But, as the number of experts becomes excessive, overlapping features among experts become unavoidable. Such redundant experts contribute minimally to overall performance, explaining the gradual deceleration in gains.

Several strategies can be adopted for optimal expert selection. While an empirical approach involves incrementally adding experts and observing the performance, a visual inspection using tools like t-SNE, as illustrated in Figure 1, can provide insights. A more advanced method leverages our generator model. Experts that generate similar samples, as determined by our generator, are deemed to have overlapping characteristics. In such cases, one of the redundant experts can be pruned from the model.

Building on our discussion, we put forth a dynamic expert adjustment strategy. Employing the $LG$, described in Section 4.2.2, we can discern experts with low utilization rates. Such underutilized experts may not significantly bolster performance and, therefore, might be considered for removal. On the other hand, when an expert's scope extends over a diverse dataset, there's an opportunity for refinement. By performing clustering on the latent vectors of the data associated with this expert, we can effectively divide the data into more distinct subsets. This allows us to allocate dedicated experts to each clustered data group, ensuring more focused and specialized learning.

## H  REPLAY METHOD ANALYSIS: RECONSTRUCTION-BASED REPLAY VS RANDOM SAMPLING

In the realm of continual learning, the efficiency of the random sampling method has been validated through numerous studies(Chaudhry et al., 2019; Vitter, 1985). In this section, we delve into an

Table 6: Comparison of performance metrics (MAE, RMSE, MAPE) between our Reconstruction-Based Replay and the random sampling strategy for replay methods.

| Replay Method | Replay Ratio | 15 min | | | 30 min | | | 60 min | | |
|---|---|---|---|---|---|---|---|---|---|---|
| | | MAE | RMSE | MAPE | MAE | RMSE | MAPE | MAE | RMSE | MAPE |
| Random | 1% | 14.29 | 23.19 | 20.78 | 16.11 | 26.43 | 22.97 | 20.66 | 33.92 | 29.22 |
| | 5% | 14.07 | 22.89 | 20.46 | 15.88 | 26.15 | 22.63 | 20.42 | 33.56 | 29.40 |
| | 10% | 13.54 | 22.05 | 19.42 | 15.25 | 25.08 | 21.45 | 19.42 | 31.84 | 27.57 |
| Ours | 1% | 12.58 | 20.63 | 16.63 | 14.12 | 23.27 | 18.40 | 17.65 | 28.95 | 22.62 |
| | 5% | 12.59 | 20.61 | 16.62 | 14.14 | 23.27 | 18.41 | 17.74 | 28.99 | 22.77 |
| | 10% | 12.61 | 20.64 | 16.63 | 14.17 | 23.29 | 18.57 | 17.82 | 29.02 | 23.01 |

analysis contrasting our Reconstruction-Based Replay methodology, as detailed in Section 4.2.4, with the random sampling. The experimental comparison is presented in Table 6. The 'Replay Ratio' in the table determines the amount of data to be replayed in proportion to the current task's graph size. Notably, our method surpasses the performance of random sampling employing over 10% replay, even when utilizing merely 1% replay. In the context of continual learning, infrequent access to pre-existing nodes from prior tasks implies a heightened efficiency, corroborating the effectiveness of our proposal. Interestingly, in our methodology, increasing the replay ratio doesn't substantially amplify performance. This suggests that our approach adeptly selects pivotal samples for the current task with minimal replay. Conversely, the performance of random sampling demonstrates a direct correlation with increased replay ratios. This can be attributed to its inherent mechanism, wherein it randomly selects samples, thereby necessitating a higher replay ratio to bolster the current task's performance.

# I  EVALUATING THE IMPACT OF SAMPLING RATIO ON MODEL PERFORMANCE

Table 7: Comparison of performance metrics (MAE, RMSE, MAPE) across varying sampling ratios.

| Sampling Ratio | 15 min | | | 30 min | | | 60 min | | | Training Time (sec) |
|---|---|---|---|---|---|---|---|---|---|---|
| | MAE | RMSE | MAPE | MAE | RMSE | MAPE | MAE | RMSE | MAPE | |
| 0% | 12.90 | 21.07 | 17.04 | 14.55 | 23.89 | 18.96 | 18.43 | 30.03 | 23.60 | 242 |
| 5% | 12.63 | 20.69 | 16.65 | 14.16 | 23.33 | 18.43 | 17.75 | 29.04 | 22.66 | 250 |
| 10% | 12.58 | 20.64 | 16.62 | 14.11 | 23.26 | 18.39 | 17.66 | 28.95 | 22.63 | 259 |
| 20% | 12.54 | 20.62 | 16.64 | 14.06 | 23.23 | 18.44 | 17.55 | 28.86 | 22.57 | 277 |
| 30% | 12.53 | 20.57 | 16.70 | 14.04 | 23.19 | 18.47 | 17.52 | 28.82 | 22.59 | 285 |

In this section, building upon the Forgetting-Resilient Sampling method we introduced in Section 4.2.3, we further investigate the influence of the sampling ratio on model performance. We tabulate our results in Table 7, where the 'Sampling Ratio' designates the proportion of data sampled relative to the current task's graph size. Throughout our experimentation, the replay ratio consistently fixed at 1%. Our empirical results underscore the significance of the sampling method. Specifically, abstaining from sampling altogether resulted in the poorest performance, whereas an optimal increase in the sampling ratio to around 10% exhibited noticeable improvements. Implementing sampling proves instrumental, as it furnishes each Expert with representative data that anchors their foundational knowledge, thereby mitigating the effects of catastrophic forgetting. The rationale behind favoring a heightened ratio for sampling over replay is twofold: unlike replay, sampled data inherently embodies and represents each Expert, and there's an inherent necessity for individualized sampling for each Expert. Additionally, our findings suggest that even with a modest sampling ratio, it's sufficient to retain the memories of each Expert, obviating the need for an excessive number of samples.

## J    EXPLORING CLUSTERING IN TRAFFIC FORECASTING AND REVISITING THE CONTRIBUTIONS OF OUR PROPOSED MODEL

In this section, we discuss the use of clustering in the general field of traffic forecasting, examining the similarities and differences with our model, and recapitulating the distinctive features or contributions of our model.

Although clustering is not a commonly used method in the field of traffic forecasting, several papers adopting it can be found. For instance, Xie et al. (2020) uses clustering to reduce computational complexity. Instead of learning from all nodes, it clusters nodes and then calculates a Pair-wise Flow Similarity Matrix for each cluster to identify the Anomalous Degree of traffic flow. Another study Guo et al. (2021) uses a method akin to Differentiable Pooling to perform Hierarchical Graph Convolution on given graphs through spectral clustering. This approach pools nodes with similar characteristics into one node to facilitate graph convolution. Ryu et al. (2022) proposes spatiotemporal correlation matrices and uses them for clustering. Following the clustering, it trains distinct prediction models for each cluster, with predictor selection being guided by mutual information. Although these methodologies share some common ground with ours in terms of employing clustering, they differ significantly in their modeling approaches, clustering strategies, prediction techniques, and fundamental concepts. Many models employ clustering for various purposes, but each does so with different objectives. Our model, in particular, has been designed to suit continual learning traffic forecasting methodologies, making it fundamentally different in both implementation and concept from existing clustering traffic forecasting models.

After exploring the broader landscape of clustering in traffic forecasting, we now turn to revisit the key contributions of our model. In the pre-training stage, we utilize k-means clustering on hidden representations derived from an AutoEncoder to form sensor groups, assigning a specialized Expert, composed of a predictor and reconstructor, to each. This concept of segmenting traffic data into multiple homogeneous groups and assigning an Expert to each minimizes interference during training and alleviates catastrophic forgetting. For making predictions, a Mixture of Experts (MOE) structure is employed, featuring a VAE-based Reconstructor to determine the appropriate Expert selection. This particular architecture, using clustering to create sensor groups and then assigning a VAE-based Reconstructor, is strategically designed to resonate with the objectives of continual traffic forecasting. Our contributions, emphasized by this strategic architecture, are highlighted by the following key aspects:

1. Our proposed 'Reconstructor-Based Knowledge Consolidation Loss' utilizes a methodology inspired by Learning Without Forgetting (LWF) Li & Hoiem (2017) to efficiently preserve knowledge gained from previous tasks while adapting to new task. This is achieved by leveraging the Reconstructor to calculate the reconstruction probabilities for sensors of the current task. These probabilities are then used to form localized groups for each Expert. This approach not only facilitates the retention of previously learned information but also aids in the smooth transition and integration of new task data.

2. Our 'Forgetting-Resilient Sampling' strategy, introduced as a supplementary approach to combat catastrophic forgetting, leverages decoders from reconstructors trained on earlier tasks to generate synthetic data samples. These synthetically produced samples effectively capture the essence of the knowledge embedded in Experts from previous tasks. By integrating these samples into the training regimen for current tasks, we enable each Expert to maintain the knowledge acquired from previous task. This methodology facilitates the generation of representative samples for each Expert, obviating the necessity of storing data from previous tasks in external memory, a significant stride in efficient memory management in continual learning frameworks.

3. A characteristic of streaming traffic networks is that even pre-existing sensors exhibit new patterns over long-term periods due to various factors like urban development. Our goal is to identify sensors unfamiliar to all Experts. Our 'Reconstruction-Based Replay' proposes an efficient method to find such nodes, i.e., universally unfamiliar to all experts, based on reconstruction probability. While existing methodologies use 10% replay to prevent catastrophic forgetting, our approach only requires 1%, showing significant performance improvements. Existing models typically need to store features from all previous year sensors in a separate memory to detect and replay sensors with similar or differing patterns between the current and previous years. In contrast, as detailed in

Section 4.2.4, our approach uses a VAE-based reconstructor, enabling us to analyze and compare with past patterns using only current task data. From a continual learning perspective, which aims to minimize past task memory usage, this presents a significant advantage.

4. Existing methodologies inevitably access pre-existing nodes' data due to the use of a subgraph structure when learning current task nodes using GNN. In contrast, the graph learning methodology we use does not employ subgraphs, thereby minimizing access to pre-existing task nodes. Furthermore, in 'Forgetting-Resilient Sampling,' synthetically generated data inherently lacks graph structural information, suggesting that generated nodes would be isolated in a predefined geographical-distance-based graph. However, the graph learning methodology we have adopted does not rely on distance-based graphs, effectively resolving this issue.

## K   ANALYSIS OF THE EFFECTS OF CLUSTERING IN THE PRE-TRAINING STAGE

Table 8: Comparison of performance between the Random-Cluster model and the TFMoE model.

| Method | 15 min | | | 30 min | | | 60 min | | |
|---|---|---|---|---|---|---|---|---|---|
| | MAE | RMSE | MAPE | MAE | RMSE | MAPE | MAE | RMSE | MAPE |
| Random-Cluster | 12.84 | 21.18 | 17.00 | 14.57 | 24.21 | 19.05 | 18.68 | 30.94 | 23.88 |
| TFMoE | 12.58 | 20.63 | 16.63 | 14.12 | 23.27 | 18.40 | 17.65 | 28.95 | 22.62 |

In this section, we explore into how clustering during the pre-training stage impacts model performance. We compare our proposed method with 'Random-Cluster' model, where nodes are randomly assigned to clusters. As shown in Table 8, our proposed model outperforms the Random-cluster model significantly. Interestingly, the Random-cluster model does not exhibit very poor performance, which leads us to discuss why our method excels and why the Random-Cluster model is somewhat effective.

First, let's delve into the Random-Cluster model. In this model, since nodes within clusters are assigned randomly, each Expert deals with a random subset of data representing the entire node distribution. Ideally, this random selection mirrors the overall node distribution. When $K$ clusters are formed using the Random-Cluster method, integrating their outcomes can be seen as a type of ensemble learning, with models trained on similar distributions that mimic the overall node distribution. As discussed in Section 1, our proposed concept involves segmenting traffic data into multiple homogeneous groups and assigning an Expert to each. Unlike the Random-Cluster model, which considers the entire data distribution, in our approach, each Expert performs predictions using specialized knowledge. When a new node is added, it is managed by the most relevant and suitable Expert, ensuring minimal impact on other unrelated Experts, which contributes to our model's effectiveness in preventing catastrophic forgetting. Furthermore, each Expert's VAE reconstructor can better generate and reconstruct the data distribution of its assigned group, and also, the predictor performs better in forecasting for the data distribution of its assigned group. This strategic design is key to why our model outperforms the Random-Cluster model.

The next interesting point is that the Random-Cluster model does not perform poorly, suggesting it also mitigates catastrophic forgetting. This raises the question: Are our proposed techniques—'Reconstruction-Based Knowledge Consolidation', 'Forgetting-Resilient Sampling', and 'Reconstruction-Based Replay'—applicable to the Random-Cluster model? The answer is 'yes'. Since the Random-Cluster model can be interpreted as an ensemble of models trained on data approximating the overall distribution, we can treat the Random-Cluster model as a single model trained on the entire data distribution for convenience. Ideally, an Expert within the Random-Cluster model should be trained to reconstruct the entire dataset, and the predictor should be trained to predict the entire dataset.

Now, let's consider whether 'Reconstruction-Based Knowledge Consolidation' can be applied to a model trained on the entire dataset. If the reconstructor is trained to reconstruct the entire dataset, applying LWF-based methods to this single model would be analogous. What about 'Forgetting-Resilient Sampling' for a model trained on the entire dataset? Using a VAE's decoder for sampling

would generate samples representing the entire data distribution of the previous task. Including these generated samples in the current task's training should evidently prevent catastrophic forgetting. Finally, can 'Reconstruction-Based Replay' be applied to a model trained on the entire dataset? Since the reconstructor is trained on the entire dataset, calculating reconstruction probability can help determine whether the single Expert can effectively handle it. Thus, the techniques we propose can be similarly employed in single models to prevent catastrophic forgetting. However, as mentioned in Section 1, we have argued that instead of using a model encompassing the entire dataset, segmenting it into multiple homogeneous groups and assigning an Expert to each is more effective. Reflecting this, the model we propose can be seen as an advanced version of this concept.

## L    COMPLEXITY ANALYSIS

Table 9: Comparison of performance between K-Means Clustering and Balanced K-Means Clustering.

| Method | 15 min | | | 30 min | | | 60 min | | |
|---|---|---|---|---|---|---|---|---|---|
| | MAE | RMSE | MAPE | MAE | RMSE | MAPE | MAE | RMSE | MAPE |
| K-Means | 12.59 | 20.65 | 16.59 | 14.14 | 23.31 | 18.51 | 17.69 | 29.13 | 22.82 |
| Blanced K-Means | 12.58 | 20.63 | 16.63 | 14.12 | 23.27 | 18.40 | 17.65 | 28.95 | 22.62 |

**Time Complexity:** The time complexity of our model is primarily dominated by the balanced k-means clustering in the Pre-Training Stage, which is $O\left(\left(N^3K + N^2K^2 + NK^3\right)\log(N + K)\right)$, and by the computational requirements of the predictor's Graph Neural Network operations in the Localized Adaptation Stage, which are $O(N^2MK)$. Here, $N$ represents the number of nodes, $K$ is the number of clusters, and $M$ denotes the diffusion step. The issue here is the substantial computational load of the balanced k-means clustering. The reason for adopting such an algorithm is the possibility of extremely skewed distributions with very few data points in the case of small datasets. Based on our experience, a minimum of about 50 nodes is required for the Expert's predictor to be trained effectively. Clusters that do not meet this criterion may not undergo proper training. However, with sufficiently large datasets where k-means clustering assigns enough data to each cluster, using conventional k-means clustering with a complexity of $O(NK)$ poses no issues. In our experiments, the performance difference between using balanced k-means clustering and conventional k-means clustering was negligible, falling within the margin of error (refer to Table 9). Therefore, the conclusion can be summarized in two scenarios:

1. **In the case of small datasets.** Employing balanced k-means clustering is beneficial for preventing the formation of clusters with a small number of data points. However, the time complexity $O\left(\left(N^3K + N^2K^2 + NK^3\right)\log(N + K)\right)$ does not lead to a significant computational overhead, as $N$ is relatively small.

2. **In the case of large datasets.** The outcomes of balanced k-means clustering and k-means clustering are similar within the margin of error. Therefore, it is sufficient to use k-means clustering with $O(NK)$ complexity, and thus the time complexity is dominated by the Graph Neural Network operations, which are $O(N^2MK)$.

If, despite using k-means clustering on large datasets, an Expert with a small number of nodes is formed, what should be done? In such a case, after completing the pre-training phase, if a cluster comprising a very small number of members is identified, this issue can be resolved by examining each cluster centroid and merging smaller clusters into the nearest sufficiently large cluster to form a sensor group.

**Space Complexity:** The space complexity of our model is predominantly governed by the $O(N^2K)$ complexity arising from the graph learning process of the predictor. The space complexity utilized by other modules apart from the predictor is $O(NK)$.

It is important to note that the calculated time complexity of $O(N^2MK)$ and space complexity of $O(N^2K)$ mainly impose a burden only on the initial task, as they utilize the entire given dataset. As we mentioned in our introduction, the reason we adopt continual learning is to prevent the forgetting of existing knowledge while training only on newly added data, rather than retraining on the entire

dataset. It is typically seen that the number of newly added nodes each year is significantly less than the number of existing nodes. After the first year, training is conducted using a number of nodes proportional to the newly added nodes, $\Delta N = |\Delta V|$. Additionally, as described in Section D, we set the diffusion step $M = 2$. Consequently, both the time complexity and space complexity after the first year become $O(\Delta N^2 K)$.

## M   LIMITATION AND FUTURE WORK

The primary limitation in our study is the scarcity of datasets in the field of continual traffic forecasting, a relatively new and not yet extensively researched area. This lack of diverse datasets makes it challenging to fully assess the generalizability of our model. Despite this limitation, we have carried out extensive visualization and experimentation to validate that our model is well-suited and effectively designed for continual traffic forecasting, and it has demonstrated significantly improved performance compared to existing models. We are optimistic that as continual traffic forecasting becomes more renowned and research in this area intensifies, there will be an introduction of a more diverse range of datasets and baselines. Such developments will not only invigorate the field but also expand opportunities for more comprehensive future studies and comparative analyses.

Our model's second limitation arises during the initial task, especially when dealing with a very large dataset. Although our approach emphasizes continual learning to avoid retraining on the entire dataset, focusing instead on new data, the initial task still requires training on the full dataset. As detailed in Section I, this leads to a complexity of $O(N^2 K)$, presenting challenges when the initial task involves a large number of nodes. In situations where similar data forms densely packed, extensive sensor groups, one feasible solution might involve sampling a suitable number of nodes within each cluster for training. In another scenario, if clustering results in numerous sensor groups, each containing a moderate amount of data, a viable approach could be to divide the initial task into several smaller subtasks. Each subtask would consist of fewer nodes, and we could apply continual learning techniques within these subtasks. The methodologies considered for applying to large datasets could potentially become new research topics within the 'Large-Scale Dataset Learning' or 'Distributed Computing' research areas. However, these extend beyond the scope of our current research, which is focused on continual learning methodologies. Therefore, we will leave these considerations as future work.

## N  ALGORITHM DESCRIPTION

---

**Algorithm 1** Traffic Forecasting Mixture of Experts (TFMoE)

---

**Require:** Training data for every task $\mathcal{D}^\tau = \{D_i^\tau\}_i = \{y_{i,t}^\tau, x_{i,t}^\tau, x_{i,w}^\tau\}_{(i,t)}$ , nodes for every task $V^\tau$, weight of clustering loss $\alpha$ and consolidation loss $\beta$, hyperparameters $n_s$ and $n_r$

1: **function** TRAIN TFMoE
2:     Pre-training reconstructor $R_k$ of each Expert using PRE-TRAININGSTAGE($\mathbf{X}_w^1 = \{x_{i,w}^1\}_i$)
3:     **for** every task $\tau$ **do**
4:         **if** $\tau = 1$ **then**
5:             Train Experts $(R_k, P_k)$ on first task data $\mathcal{D}^1$ with loss $\mathcal{L} = \mathcal{L}_\mathcal{O} - \beta\mathcal{L}_{ELBO}^{SG}$
6:         **else**
7:             Synthetic data samples $D^s \leftarrow$ FORGETTINGRESILIENTSAMPLING($\phi^{(\tau-1)}, n_s$)
8:             Important nodes $V_R \leftarrow$ RECONSTRUCTIONBASEDREPLAY($V^{(\tau-1)}, \phi^{(\tau-1)}, n_r$)
9:             Aggregate data for current task: $D^* \leftarrow D^s \cup \{D_i^\tau \mid i \in \Delta V^\tau \cup V_R\}$
10:            Construct $LG_k$ on $D^*$
11:            Train Experts $(R_k, P_k)$ on aggregated data $D^*$ with loss $\mathcal{L} = \mathcal{L}_\mathcal{O} - \beta\mathcal{L}_{ELBO}^{LG}$
12:        **end if**
13:    **end for**
14: **end function**

---

**Algorithm 2** PRE-TRAINING STAGE

---

**Require:** First task data $\mathbf{X}_w^1$

1: Train autoencoder $R_{pre}$ with loss $\mathcal{L}_{recon}$
2: Get $K$ cluster centroids $[\mu_1; \ldots; \mu_K]$ using latent vectors $\kappa_i = R_{pre;enc}(x_{i,w}^1)$
3: Train autoencoder $R_{pre}$ with loss $\mathcal{L}_p = \mathcal{L}_{recon} + \alpha\mathcal{L}_{cluster}$
4: Construct $SG_k$ utilizing cluster assignment probability
5: Train each reconstructor $R_k$ with loss $\mathcal{L}_{ELBO}^{SG}$ based on cluster $SG_k$
6: **return** pre-trained reconstructor $R_k$

---

**Algorithm 3** FORGETTING-RESILIENT SAMPLING

---

**Require:** $\phi^{(\tau-1)}, n_s$

1: **for** each Expert $k$ **do**
2:     Sample latent variables: $z_{k,i} \sim p(z_k; \phi_k^{(\tau-1)})$ for $i = 1$ to $n_s/k$
3:     Generate data samples: $x_{w_{k,i}} \sim p(x_w|z_{k,i}; \phi_k)$
4:     Aggregate samples: $X_k^s \leftarrow \{x_{w_{k,1}}, x_{w_{k,2}}, \ldots, x_{w_{k,n_s/k}}\}$
5: **end for**
6: Aggregated dataset: $X^s \leftarrow \bigcup_{k=1}^K X_k^s$
7: Synchronize aggregated dataset: $D^s \leftarrow \text{Sync}(X^s)$
8: **return** $D^s$

---

**Algorithm 4** RECONSTRUCTION-BASED REPLAY

---

**Require:** $V^{(\tau-1)}, \phi^{(\tau-1)}, n_r$

1: Rank nodes by reconstruction probability $\Sigma_k \log p(x_{i,w}; \phi_k^{(\tau-1)})$ in ascending order
2: Select the first $n_r$ nodes to form $V_R$ from node set $V^{(\tau-1)}$
3: **return** $V_R$

---

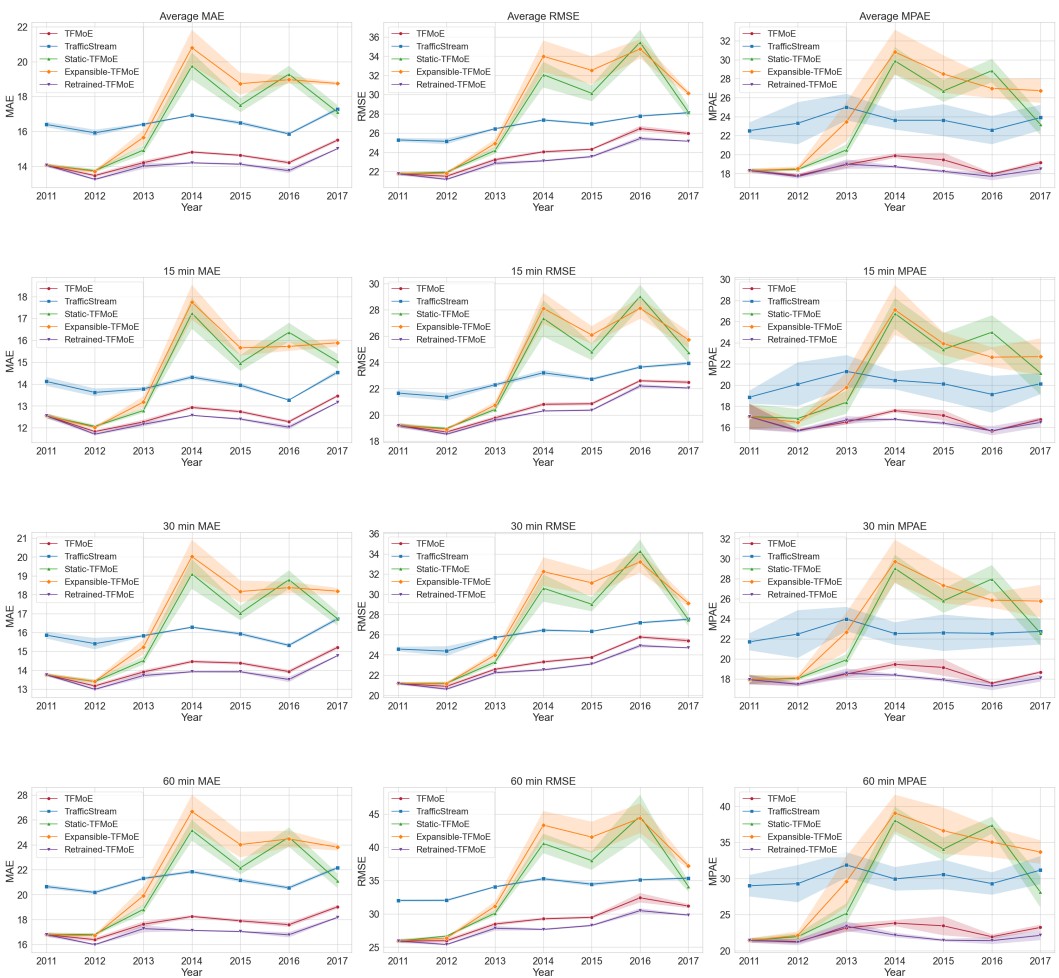

Figure 9: The MAE, RMSE, and MAPE of traffic flow forecasting over consecutive years.

