# OpenReview forum: "Continual Traffic Forecasting via Mixture of Experts"
_ICLR.cc/2024/Conference — Submitted to ICLR 2024_

### Official Review · Reviewer_p8v4 · 2023-10-23

**Soundness:** 3 good
**Presentation:** 2 fair
**Contribution:** 3 good
**Rating:** 6
**Confidence:** 1

**Summary:**

In this paper, the authors focus on one specific traffic forecasting problem where the traffic information expands over time, and they aim to design a method that can simultaneously learn from the new data as well as the old data. Instead of training the model on the entire dataset once new data arrives, they adapt continual learning where the model can integrate the knowledge of the new data while not forgetting the past knowledge. Furthermore, they propose a set of modules to improve the performance of the model, such as clustering, VAE-based reconstruction, and forgetting-resilient sampling. The experiments on real-world datasets show the performance of the proposed method.

**Strengths:**

1. It proposes a novel approach to address the challenges of catastrophic forgetting and inefficiency in traffic forecasting under evolving networks. The Traffic Forecasting Mixture of Experts (TFMoE) method segments traffic flow into multiple homogeneous groups and assigns expert models to specific patterns, achieving superior performance and resilience in long-term streaming network datasets.
2. The paper provides extensive experimental results on a real-world long-term streaming network dataset, demonstrating the effectiveness and efficiency of TFMoE.
3. The paper emphasizes the importance of ethical considerations and reproducibility in scientific research, providing an ethics statement and reproducibility statement.

**Weaknesses:**

1. The Structure of the paper could be improved - Section 4 seems to be a bit too long, while part of the experiments, especially the settings and principles have to be left in the appendix. As I am not an expert in this area, section 4 is a little hard to follow, many complex modules are introduced in this section which makes it easy to lose.
2. As far as I know, clustering seems to be a common technique in the field of machine learning, while seldom reference is about the utilization of clustering in the traffic area. I think a discussion of clustering in this area is needed. Besides, the clustering operation seems to cost extra resources, in the sense of time or memory. Do the authors consider that?
3. The paper does not discuss the generalizability of the proposed method to other traffic datasets, as there is only one dataset in the paper. I understand the hard of acquiring real-world data. Can we just use the real-world roadnet but generate several new traffic flows? In this way we can evaluate the methods on multiple datasets and see its generalization ability.

**Questions:**

As I mentioned in the weakness, I suggest the authors discuss more about clustering in the literature and try to improve their experiments.

---

> ### Author Response · Authors · 2023-11-14
> **Response to Reviewer p8v4**
>
> We appreciate the time and effort you have dedicated to reviewing our paper. Your detailed comments have helped us to enhance our work significantly.
>
> W1: Firstly, I apologize for any inconvenience caused while reading the paper. The contents of Section 4 are considered essential components of our model, which made it challenging to simplify the content. To aid in understanding our model we have added various sections and discussions in the appendix for a more detailed explanation of the model's architecture. Please refer to the updated paper for this information.
>
> W2, Q1: Thank you for your constructive advice. In response to your comments, we have created a new section titled 'Appendix J: Exploring Clustering in Traffic Forecasting and Revisiting the Contributions of Our Proposed Model.' In this section, we examine how clustering methodologies are used in existing traffic forecasting, their purposes, and how our modeling aligns or differs from these methods. We also agree that the time and space complexity of the clustering operation is an important consideration. Reflecting this, we have added 'Appendix L: Complexity Analysis,' where we analyze the time and space complexity of the entire model, including the clustering operation. Please refer to the updated paper for this information.
>
> W3: Thank you for your valuable feedback. One limitation of our research, as you pointed out, is its generalization ability. The scarcity of datasets in the field of continual traffic forecasting, a relatively new and under-researched area, poses challenges in fully assessing the generalizability of our model. The dataset we currently use represents a real-world long-term streaming network, encompassing a lengthy period of seven years with new sensors added annually. Our model architecture is specifically tailored for such long-term streaming networks. If we were to create a new dataset suitable for long-term streaming networks, it would likely be challenging to do so quickly in a short period. Most existing datasets used in traffic forecasting are collected over short durations and thus are not directly applicable for evaluating our model tailored for long-term streaming networks. Despite these challenges, we have conducted extensive visualization and experimentation to demonstrate that our model is well-adapted and effectively designed for continual traffic forecasting, exhibiting significant performance improvements over existing models. We are hopeful that as continual traffic forecasting becomes more recognized and researched, a wider variety of datasets and benchmarks will become available, energizing the field and enabling more comprehensive future studies and comparative analyses. This discussion will be included in the newly created 'Appendix M: Limitation and Future Work' section.

---

> > ### Comment · Reviewer_p8v4 · 2023-11-20
> >
> > Thank you very much for the detailed response to address my concerns, but I am still a little concerned about the dataset problem. As I mentioned in my initial review,  I am not an expert in this area. Therefore, in my standard of common machine learning papers, tests on multiple datasets are necessary to verify the performance of the proposed model. I have increased my score to 6 but with less confidence. For now, I’m neutral about the paper and wouldn’t mind either accepting or rejecting it.

---

### Official Review · Reviewer_TPK5 · 2023-10-29

**Soundness:** 3 good
**Presentation:** 2 fair
**Contribution:** 2 fair
**Rating:** 3
**Confidence:** 4

**Summary:**

The manuscript delineates a novel approach, termed as TFMoE, which is devised for continual traffic forecasting where the traffic patterns continually evolve over time. In addressing this issue and the challenge of catastrophic forgetting, the TFMoE exhibits an innovative usage of Mixture-of-Experts,  along with three complementary mechanisms - namely Reconstructor-Based Knowledge Consolidation Loss, Forgetting-Resilient Sampling, and Reconstruction-Based Replay, that endow the model with superior performance in comparison to baseline methods.

**Strengths:**

S1. The paper's significance is underscored by its goal to address a real-world problem.

S2.  The TFMoE model's novelty is encapsulated in its innovative usage of Mixture-of-Experts,  along with three complementary mechanisms, effectively addressing the unique challenges in continual traffic forecasting.

S3. The evaluation is comprehensive, with comparisons to baseline methods providing a compelling demonstration of the superior performance of the TFMoE model.

S4. The clarity of the manuscript enhances accessibility for readers, facilitating a straightforward understanding of the proposed approach.

**Weaknesses:**

W1.  Although the paper provides a comprehensive explanation of the methodology, further technical insights regarding the implementation and specific algorithms within the TFMoE method would be beneficial.

W2. The paper falls short in providing a detailed analysis of the limitations of the proposed TFMoE, a factor which could be significant for future research and practical applications.

W3. The computational complexity of the TFMoE algorithm, especially in the phases of continual training and forecasting, which could be a concern for large-scale datasets, is not discussed in the manuscript.

W4.  Although the paper employs sound-good methodology and achieves competitive performance,  further efforts regarding the technical innovation and methodological novelty would be beneficial.

W5. The manuscript could delve deeper into the parameter study, a factor which could be pivotal for practical applications.

W6. A more detailed exposition of the dataset and the experimental settings used in the evaluation, including their characteristics and potential biases, would enrich the manuscript.

**Questions:**

C1. How does the proposed method address the heterogeneous graph structures over time?

C2. Does the model's performance have a dependence on the pre-trained clustering results?

C3. Does the model have sensitivity to the parameters?

---

> ### Author Response · Authors · 2023-11-14
> **Response to Reviewer TPK5**
>
> We appreciate the time and effort you have dedicated to reviewing our paper. Your detailed comments have helped us to enhance our work significantly.
>
> W1, W4: I agree with your opinion that, although technical insights regarding each methodology are provided in the Introduction and Proposed Method sections of the paper, further efforts to clarify these points are necessary. In response, we have created a new section titled ‘Appendix J: Exploring Clustering in Traffic Forecasting and Revisiting the Contributions of Our Proposed Model’ to reorganize and clarify our contributions. Please refer to the updated paper for this information.
>
> W2: In response to your feedback, we have created a new section in the appendix titled 'Appendix M: Limitation and Future Work' where we analyze the current limitations of our model. Please refer to the updated paper for this information.
>
> W3: Thank you for the valuable feedback. In response, we have added a new section titled 'Appendix L: Complexity Analysis' where we conduct an analysis of both time complexity and space complexity. As mentioned in the Introduction, the reason for using continual learning is that retraining the entire network is inefficient. Instead, we use continual learning methods to maintain high performance while training only a few newly added nodes. However, in our methodology, when we train the first task, we use the entire dataset, which can be problematic. We have addressed this discussion in a newly created section 'Appendix M: Limitation and Future Work'. Please refer to the updated paper for this information.
>
> W5, C3: We have presented experimental results for the core hyperparameters that we believe are crucial in our modeling. Experiments on the number of Experts can be found in the appendix 'Appendix G: Optimizing Expert Selection.' Experiments on the replay ratio for 'Reconstruction-Based Replay' are available in 'Appendix H: Replay Method Analysis: Reconstruction-Based Replay vs Random Sampling.' Experiments on the sampling ratio for 'Forgetting-Resilient Sampling' are detailed in 'Appendix I: Evaluating the Impact of Sampling Ratio on Model Performance.' However, as you suggest, experiments on the sensitivity of various other hyperparameters are also important. We plan to add a section related to parameter sensitivity and will update it for inclusion in the final version of the paper.
>
> W6: We have described the dataset in a manner that is typical in the field of traffic forecasting. This can be verified in 'Appendix A: Dataset.' Additionally, details about the experimental settings, such as dataset preprocessing, can be found in 'Appendix C: Evaluation Protocol.' Information on the model's hyperparameter settings is available in 'Appendix D: Implementation Details.' Please refer to these sections for more information.
>
> C1: In our research, we focus on a homogeneous graph structure that aligns with the nature of our 1-dimensional traffic flow data. Each node in our graph represents a data point in traffic flow, and our structure does not incorporate heterogeneous elements. The primary aim of our study is to model the changes in traffic flow over time, capturing the patterns and variations within this specific context. As such, the question of addressing heterogeneous graph structures does not directly apply to our work, since our methodology and data are inherently homogeneous. While our current approach is specialized for uniform data types, we recognize the potential for future research to explore and incorporate more complex, heterogeneous graph structures, expanding the scope and applicability of our work.
>
> C2: If 'pre-trained clustering' refers to the clustering methodology in the pre-training stage, then such an experiment would indeed be a good way to support our main idea. We appreciate the suggestion and have created a new section titled 'Appendix K: Analysis of the Effects of Clustering in the Pre-training Stage' in response to your comments. In this section, we have conducted experiments and analyzed the results. Please refer to the updated paper for this information.

---

### Official Review · Reviewer_WWUg · 2023-11-19

**Soundness:** 2 fair
**Presentation:** 2 fair
**Contribution:** 2 fair
**Rating:** 3
**Confidence:** 3

**Summary:**

The paper tries to handle the catastrophic forgetting problem in the fields of traffic forecasting. The authors introduce the method referred to as TFMoE, which is based on mixture of experts technique.

**Strengths:**

1. The authors decompose the problem in structured way
2. The paper is well written and easy to follow
3. The problem which this paper handles is very interesting

**Weaknesses:**

1. Experiments are only done in one dataset. It is better to extend the scope of experiments to validate the method.
2. While the components are well combined, most of them are existing techniques.
3. Lack of analysis. Only the ablation of components were conducted. The paper would benefit from the additional analysis.

**Questions:**

See weakness

---

> ### Author Response · Authors · 2023-11-21
> **Response to Reviewer WWUg**
>
> We appreciate the time and effort you have dedicated to reviewing our paper. Your detailed comments have helped us to enhance our work significantly.
>
> W1: As you have mentioned, our research was conducted on a single dataset. Continual traffic forecasting is a relatively new research field with limited datasets available and a scarcity of baselines. We acknowledge these limitations and have discussed them in the 'Appendix M: Limitation and Future Work' section of our paper. We also address this topic in our response to reviewer p8v4, which we invite you to consult for further details. To mitigate these issues, we have conducted a variety of experiments and visualizations for every component and module proposed in our study. We hope this approach will be considered in your assessment of our work.
>
>
> W2: If by 'existing techniques' you are referring to methods like Clustering, Mixture of Experts (MoE), VAE sampling, Learning without Forgetting (LwF), and Graph Learning that we employed to prevent catastrophic forgetting in continual traffic forecasting, then indeed, these are established methods, and we have duly cited them in our paper. The novelty of our work lies in the exploration and analysis of how these methodologies can be appropriately and effectively adapted to the field of continual traffic forecasting. From this perspective, if you are aware of any other papers employing similar techniques in this context, we would greatly appreciate your sharing them with us. This would further enrich our understanding and the scope of our research.
>
>
> W3: Firstly, we conducted various ablation studies to demonstrate the efficiency and validity of each component in our model. Our analysis goes beyond mere ablation; in 'Appendix E: Analysis of Reconstructor and Predictor Outputs' and 'Appendix F: Visualization of Synthetic Data Generated through Decoders', we provided visualizations of our model. Furthermore, in 'Appendix H: Replay Method Analysis: Reconstruction-Based Replay vs Random Sampling', we showcased the effectiveness of our methods through comparative analysis. In response to feedback from other reviewers, our updated version of the paper includes 'Appendix K: Analysis of the Effects of Clustering in the Pre-training Stage', which presents experiments and analysis on the impact of clustering during the pre-training stage.
>
> Given that your review was submitted towards the end of the Discussion period, it may not be feasible to immediately incorporate your feedback. However, we are very open to your suggestions. If you could specify what content you recommend for inclusion in the additional analysis, we will make effort to incorporate it into the final version of our paper.

---

### Meta-Review · Area_Chair_wxgN · 2023-12-06

**Metareview:**

The reviewers of Submission 6704, notably Reviewer TPK5  and Reviewer p8v4, have provided their assessments of the paper, which introduces the TFMoE method for continual traffic forecasting. While acknowledging the novel approach and strengths of the paper, they also pointed out several weaknesses and raised questions that led to their overall recommendation of not accepting the paper for publication in its current state.

Reviewer TPK5 appreciated the innovative aspects of TFMoE and its comprehensive evaluation but expressed concerns about the lack of technical insights, the absence of a detailed analysis of the method's limitations, the unaddressed computational complexity, and insufficient exploration of the method's technical innovation and parameter study. The reviewer also noted the need for a more detailed explanation of the dataset and experimental settings. Questions were raised about the model's handling of heterogeneous graph structures, its dependency on pre-trained clustering results, and its sensitivity to parameters. The final rating given was a rejection, citing that the paper is not good enough for acceptance.

Reviewer p8v4 commended the proposal of a novel approach and the extensive experimental results, but pointed out weaknesses in the paper's structure, a lack of discussion on the utilization of clustering in traffic forecasting, and concerns about the generalizability of the method due to the use of only one dataset. They suggested further discussion on clustering in the literature and improving experiments. The reviewer's final stance was neutral, with an increased score but less confidence.

In response to these reviews, the authors made revisions and addressed most of the concerns, including adding new sections in the appendix to clarify and enhance the content of the paper. Despite these efforts, the concerns about the dataset and the need for tests on multiple datasets remained a significant point of contention for Reviewer p8v4.

**Justification For Why Not Higher Score:**

While the paper presents a novel method and has undergone revisions to address the reviewers' concerns, its current form still lacks in areas such as detailed technical insights, comprehensive analysis of limitations, discussion of computational complexity, generalizability, and thoroughness in the parameter study. These shortcomings contribute to the recommendation against its acceptance for publication at this time.

**Justification For Why Not Lower Score:**

N/A

---

### Decision · Program_Chairs · 2024-01-16

Reject